# Learning to stand with unexpected sensorimotor delays

Brandon G Rasman[1,2,3], Patrick A Forbes[2], Ryan M Peters[3,4,5], Oscar Ortiz[3,6], Ian Franks[3], J Timothy Inglis[3,7], Romeo Chua[3], Jean-Sébastien Blouin[3,7,8]*

[1]School of Physical Education, Sport, and Exercise Sciences, University of Otago, Dunedin, New Zealand; [2]Department of Neuroscience, Erasmus MC, University Medical Center Rotterdam, Rotterdam, Netherlands; [3]School of Kinesiology, University of British Columbia, Vancouver, Canada; [4]Faculty of Kinesiology, University of Calgary, Calgary, Canada; [5]Hotchkiss Brain Institute, Calgary, Canada; [6]Faculty of Kinesiology, University of New Brunswick, Fredericton, Canada; [7]Djavad Mowafaghian Centre for Brain Health, University of British Columbia, Vancouver, Canada; [8]Institute for Computing, Information and Cognitive Systems, University of British Columbia, Vancouver, Canada

**ABSTRACT** Human standing balance relies on self-motion estimates that are used by the nervous system to detect unexpected movements and enable corrective responses and adaptations in control. These estimates must accommodate for inherent delays in sensory and motor pathways. Here, we used a robotic system to simulate human standing about the ankles in the antero-posterior direction and impose sensorimotor delays into the control of balance. Imposed delays destabilized standing, but through training, participants adapted and re-learned to balance with the delays. Before training, imposed delays attenuated vestibular contributions to balance and triggered perceptions of unexpected standing motion, suggesting increased uncertainty in the internal self-motion estimates. After training, vestibular contributions partially returned to baseline levels and larger delays were needed to evoke perceptions of unexpected standing motion. Through learning, the nervous system accommodates balance sensorimotor delays by causally linking whole-body sensory feedback (initially interpreted as imposed motion) to self-generated balance motor commands.

*For correspondence: jsblouin@mail.ubc.ca

Competing interest: The authors declare that no competing interests exist.

## Introduction

The nervous system learns and maintains motor skills by forming probabilistic estimates of self-motion. The resulting inferred relationships between sensory and motor signals form a representation of the world and self that allows the brain to identify unexpected behavior and adapt motor control (*Friston, 2010*; *Krakauer and Mazzoni, 2011*; *Wolpert et al., 2011*). Due to neural conduction delays, these estimates of self-motion rely on the expected timing between motor commands and resulting sensory feedback. As such, errors associated with self-generated movement increase with larger feedback delays (*Gifford and Lyman, 1967*; *Miall et al., 1985*; *Smith et al., 1960*). Through repeated exposure to an imposed delay, the brain can learn to expect the delayed feedback associated with self-motion, leading to improvements in movement control with delays up to 430 ms (*Cunningham et al., 2001*; *Miall and Jackson, 2006*). When balancing upright, sensory feedback associated with lower-limb motor commands is delayed by up to ~100–160 ms (*Forbes et al., 2018*; *Kuo, 2005*; *van der Kooij et al., 1999*). As a consequence of these relatively long delays, computational feedback models of upright standing predict that balance controllers cannot adjust their sensorimotor gains and stabilize balance in the anteroposterior (AP) direction with imposed delays larger than ~300–340 ms

**eLife digest** When standing, neurons in the brain send signals to skeletal muscles so we can adjust our movements to stay upright based on the requirements from the surrounding environment. The long nerves needed to connect our brain, muscles and sensors lead to considerable time delays (up to 160 milliseconds) between sensing the environment and the generation of balance-correcting motor signals. Such delays must be accounted for by the brain so it can adjust how it regulates balance and compensates for unexpected movements.

Aging and neurological disorders can lead to lengthened neural delays, which may result in poorer balance. Computer modeling suggests that we cannot maintain upright balance if delays are longer than 300-340 milliseconds. Directly assessing the destabilizing effects of increased delays in human volunteers can reveal how capable the brain is at adapting to this neurological change.

Using a custom-designed robotic balance simulator, Rasman et al. tested whether healthy volunteers could learn to balance with delays longer than the predicted 300-340 millisecond limit. In a series of experiments, 46 healthy participants stood on the balance simulator which recreates the physical sensations and neural signals for balancing upright based on a computer-driven virtual reality. This unique device enabled Rasman et al. to artificially impose delays by increasing the time between the generation of motor signals and resulting whole-body motion.

The experiments showed that lengthening the delay between motor signals and whole-body motion destabilized upright standing, decreased sensory contributions to balance and led to perceptions of unexpected movements. Over five days of training on the robotic balance simulator, participants regained their ability to balance, which was accompanied by recovered sensory contributions and perceptions of expected standing, despite the imposed delays. When a subset of participants was tested three months later, they were still able to compensate for the increased delay.

The experiments show that the human brain can learn to overcome delays up to 560 milliseconds in the control of balance. This discovery may have important implications for people who develop balance problems because of older age or neurologic diseases like multiple sclerosis. It is possible that robot-assisted training therapies, like the one in this study, could help people overcome their balance impairments.

(*Milton and Insperger, 2019*; *van der Kooij and Peterka, 2011*). These predictions contrast the reported upper-limb sensorimotor adaptation to imposed delays (*Cunningham et al., 2001*; *Miall and Jackson, 2006*). The present study aims to directly quantify the destabilizing effects of imposed delays between ankle torque and whole-body motion during standing balance, and to determine the underlying mechanisms responsible for any subsequent adaptation and learning.

Imposed delays inserted within the balance control task are expected to increase postural oscillations and, past a critical delay, lead to falls (*Bingham et al., 2011*; *Milton and Insperger, 2019*; *van der Kooij and Peterka, 2011*). The sensorimotor mechanisms underlying these predicted effects, however, are unknown. Of particular interest is the vestibular control of balance due to its task-dependent modulations (*Fitzpatrick and McCloskey, 1994*; *Forbes et al., 2016*; *Luu et al., 2012*; *Mian and Day, 2014*), which rely on predictable associations between self-generated motor and resulting sensory signals. For example, participants exposed to novel vestibular feedback of balance motion initially exhibit increased postural oscillations but decreased muscle responses to a vestibular error signal (*Héroux et al., 2015*). With practice, participants improve their balance and vestibular-evoked muscle responses return to baseline amplitudes, suggesting that the brain updated its vestibular estimates of self-motion. Based on these observations, we hypothesized that increasing balance delays would initially increase whole-body motion and attenuate vestibular-evoked responses but these effects would diminish following a learning period. Another critical feature of probabilistic associations between motor and sensory cues is our ability to perceptually distinguish between self-generated or externally imposed motions. Imposed sensorimotor delays during self-generated movements evoke a sensation interpreted to arise from external causes rather than oneself (*Blakemore et al., 1999*; *Farrer et al., 2008*; *Wen, 2019*). Repeated exposures to delayed self-generated touch can re-align the perceived timing of the contact with the imposed delay (*Kilteni et al., 2019*; *Stetson et al., 2006*). Therefore, we further hypothesized that balance behavior under imposed delays would

be inferred as externally imposed motion but this likelihood would decrease through repeated exposure to the delay.

To test these hypotheses, we performed three experiments where participants balanced in a robotic simulator (*Figure 1*) in the AP direction with imposed delays ranging from 20 to 500 ms. These delays were in addition to the physiological delays (~100–160 ms) inherent to standing balance. In Experiment 1, we characterized standing behavior across this range of delays. Generally, whole-body sway variability increased with larger imposed delays and participants repeatedly fell into virtual limits of the balance simulation (i.e., 6° anterior and 3° posterior) for added delays larger than ~200–300 ms. In Experiment 2, participants trained to balance upright with a 400 ms added delay (testing beyond the critical delay previously proposed) for 100 min over five consecutive days. We probed the vestibular-evoked muscle responses and the perception of body motion before, after and 3 months following training to assess how the brain adapted to and processed the delayed sensory feedback. Initially, participants exhibited increased postural oscillations while the vestibular contributions to balance decreased and their perception of unexpected balance motion increased. After training, participants' balance behavior improved, their vestibular-evoked responses increased, and larger imposed delays were needed to elicit perceptions of unexpected balance motion. To further evaluate the effect of imposed delays on the vestibular control and perception of balance, we exposed participants to transient delays (Experiment 3). Within a few seconds of transitioning to a 200 ms delay, whole-body sway variability increased, vestibular responses attenuated (~70–90% decline) and participants perceived unexpected balancing motion. Collectively, our findings demonstrate how novel sensorimotor delays disrupt standing balance and suggest that the nervous system can learn to maintain standing balance with imposed delays by associating delayed whole-body motion with self-generated balancing motor commands.

## Results

### Experiment 1: imposed sensorimotor delays increase postural oscillations

Thirteen healthy participants were instructed to stand quietly on a robotic balance simulator for 60 s trials (Materials and methods) while experiencing fixed imposed delays (20, 100, 200, 300, 400, 500 ms) between the torques generated at their feet and resulting whole-body motion. These delays were in addition to the ~100–160 ms sensorimotor feedback delays inherent to standing balance. Whole-body sway was recorded throughout these trials to quantify the effect of the additional delay on standing balance. The robot was programmed to rotate the whole body in the AP direction about the participant's ankles. Angular position limits of 6° anterior and 3° posterior from vertical were imposed into the simulation to represent the physical limits of sway during standing balance, whereby the robot constrained the angular rotation when these virtual limits were exceeded (see Materials and methods).

While balancing on the robotic simulator at the 20 ms delay (baseline condition), all participants maintained standing balance with small postural oscillations around their preferred upright posture (sway velocity variance: $0.07 \pm 0.07$ [°/s]$^2$ [mean ± standard deviation]). Whole-body oscillations increased with the imposed delays, leading to marked difficulties in maintaining a stable posture when a 400 ms delay was imposed. Representative data (*Figure 2A*) illustrate a participant exceeding the virtual balance limits (i.e., whole-body position traces exceeding dashed lines) 20 times within a 60 s period. This observation was confirmed in the group data. No participant exceeded these virtual balance limits at the 20 ms condition (only one participant reached the limit during the 100 ms condition) whereas every participant exceeded the virtual limits at least once within the 60 s balance period when delays were ≥200 ms. There was a main effect of delay on sway velocity variance (extracted over 2 s windows of continuous balance; Materials and methods), such that sway velocity variance was smallest for the 20 ms condition ($0.07 \pm 0.07$ [°/s]$^2$) and increased with the magnitude of the imposed delay and reached a maximum at 400 ms ($21.08 \pm 15.41$ [°/s]$^2$; p<0.001; *Table 1* and *Figure 2B*). We also quantified the percent time participants balanced within the virtual limits. There was a main effect of delay on the percent time within the balance limits (p<0.001; *Table 1* and *Figure 2B*), which decreased from 100% ± 0 % during the 20 ms condition to 54% ± 9 % during the 500 ms condition. Decomposition of the main effects revealed that participants exhibited greater sway velocity variance

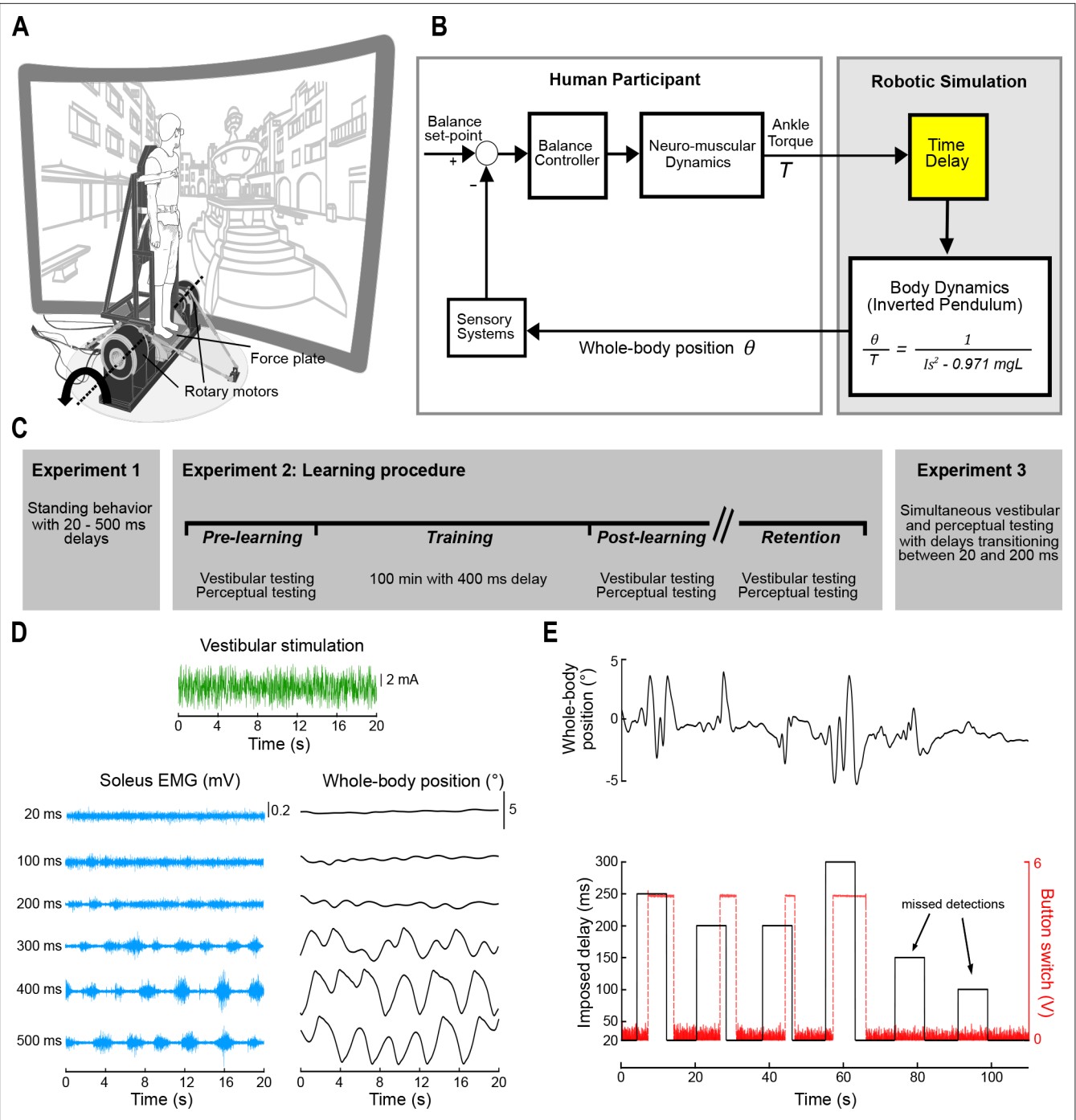

**Figure 1.** Experimental setup and block diagram of robotic simulation. (**A**) The participant stood on a force plate mounted to an ankle-tilt platform and was securely strapped to a rigid backboard. The ankle-tilt platform and backboard were independently controlled by rotary motors. In all experiments, the ankle-tilt platform was held at horizontal (earth-fixed reference) while the backboard rotated the participant in the anteroposterior plane. Motion of the backboard was controlled by ankle torques exerted on the force plate based on the mechanics of an inverted pendulum. The backboard rotated about an axis that passed through the participant's ankles (dashed line). Participants wore 3D goggles and viewed a virtual scene of a courtyard. (**B**) Participants balanced the robotic simulator as it operated with a 20–500 ms delay. Torque signals (*T*) from the force plate were buffered in the robotic simulation computer model such that angular rotation of the whole body (*θ*) about the ankle joint could be delayed. (**C**) Experimental design. Experiment 1 involved testing standing balance when naïve participants (n = 13) were first exposed to delays. Experiment 2 involved learning to balance with delays and was performed in two groups: vestibular testing and perceptual testing (*see Experiment 2 methods*). All participants who performed the learning experiments (vestibular testing group, n = 8; perceptual testing group, n = 8) completed an identical training protocol. The vestibular and perceptual tests were completed before, immediately after, and ~3 months following training. Training was completed over 5 days, in which the

*Figure 1 continued on next page*

*Figure 1 continued*

participant balanced the robotic simulator with a 400 ms delay (20 min per day). Experiment 3 tested a new group of participants (n = 7) and evaluated the time-dependent attenuation in vestibular-evoked responses together with changes in sway behavior and perception of unexpected balance motion. Trials in Experiment 3 were of similar design to perceptual testing in Experiment 2 (see panel E), except that the robot only transitioned between baseline (20 ms) and 200 ms delays. (**D**) Raw data of a sample participant from Experiment 2 vestibular testing. The participant was exposed to electrical vestibular stimulation while balancing the robotic simulator as it operated at fixed delays. Raw traces of the vestibular stimulus (green), soleus muscle EMG (blue), and whole-body position (black) are shown for a single trial at each delay condition. (**E**) During perceptual testing (Experiments 2 and 3), the participant balanced the robotic balance simulator and held a button switch. Delays were manipulated in the robotic balance simulation and the participant was required to press and hold the button when unexpected balance motion was detected. Raw data traces of whole-body position (black, upper trace), imposed simulation delay (black, lower trace), and the button switch (red) are shown during a perceptual trial from Experiment 2. Black arrows indicate examples of imposed delays that did not elicit a perceptual detection. *Figure 1A* was adapted from *Shepherd, 2014*.

and lower percentage of trial duration within the virtual limits compared to the 20 ms condition when imposed delays were ≥200 ms (all p-values <0.05).

## Experiment 2: learning to stand upright with a 400 ms delay

In a second set of experiments, we tested whether humans can adapt and learn to stand with imposed sensorimotor delays. Participants (n = 16) performed a training protocol over five consecutive days (two 10 min trials per day) where they balanced on the robot with a 400 ms delay. To explore the neural processes involved in balancing with novel sensorimotor delays, we characterized the participants' vestibular control of balance (vestibular testing, see below) or their perceptual detection of unexpected motion (perceptual testing, see below) before and after training. Twelve participants also returned ~3 months later to examine whether any learning was retained.

Within the first minute of training with the 400 ms delay, no participant could remain upright: on average, they reached the forward or backward virtual balancing limits 18 ± 5 times (see representative participant in *Figure 3A*) and could only remain within the balancing limits for 64% ± 9 % of the time (or 38.5 s). This unstable balancing behavior was characterized by large whole-body sway velocity variance (12.62 ± 9.03 $[°/s]^2$). During training, participants progressively reduced the variance of their sway velocity and increased the percentage of time they balanced within the virtual limits. The first minute of each day (i.e., start of every 20 min interval) was characterized by an increase of sway velocity variance and a decrease of percentage of time within the limits relative to the last min of the previous day (see filled circles, *Figure 3B*). By the end of training (100 min), participants exhibited an ~80 % decrease in sway velocity variance and a 51 % increase in percent time within the virtual limits. First-order exponential fits estimated the changes in sway velocity variance and percent time within the limits. The time constant for the decrease in sway velocity variance (i.e., 63.2 % attenuation) was 27.9 min (corresponding to a value of 6.44 $[°/s]^2$), and the time constant for the increase in percent time within limits (i.e., 63.2 % increase) was 32.5 min (corresponding to a value of 85 % within the balance limits). By the last 60 s of training, participants could balance the robot within the simulation limits on average for 97% ± 3 % of the time (or 59.2 s), with four participants capable of balancing for the final 60 s interval without reaching a limit. However, the smallest sway velocity variance observed with a 400 ms imposed delay remained ~38× greater than the baseline condition (400 ms at 93rd min vs. 20 ms variance: 1.91 ± 1.12 $[°/s]^2$ vs. 0.05 ± 0.05 $[°/s]^2$; $t_{(15)}$ = 6.74: p<0.001).

When participants (n = 12) returned for retention testing ~3 months later, these balance improvements were partially maintained. Sway velocity variance in the first minute of retention testing was ~60.8 % lower than the sway velocity variance from the first minute of training (4.95 ± 2.32 $[°/s]^2$ vs. 12.62 ± 9.03 $[°/s]^2$; independent samples t-test: $t_{(26)}$ = –2.86, p<0.01). Sway velocity variance at the first minute of retention testing, however, remained greater than the last minute of training (4.95 ± 2.32 $[°/s]^2$ vs. 2.55 ± 1.76 $[°/s]^2$; independent samples t-test: $t_{(26)}$ = 3.11, p<0.01). Similarly, the first minute of retention was associated with a greater percentage of time within the balancing limits compared to the first minute of training (88% ± 9% vs. 64% ± 9%; independent samples t-test: $t_{(26)}$ = 6.67, p<0.001), but less than the last minute of training (88% ± 9% vs. 97% ± 3%; independent samples t-test: $t_{(26)}$ = –3.68, p<0.01). When using only data from participants who performed the retention session (n = 12; paired t-tests with df = 11), sway velocity variance and percent time within the balance limits revealed identical results (all p-values < 0.01). Overall, these results indicate that while standing with an imposed 400 ms delay is initially difficult (if not impossible), participants learn

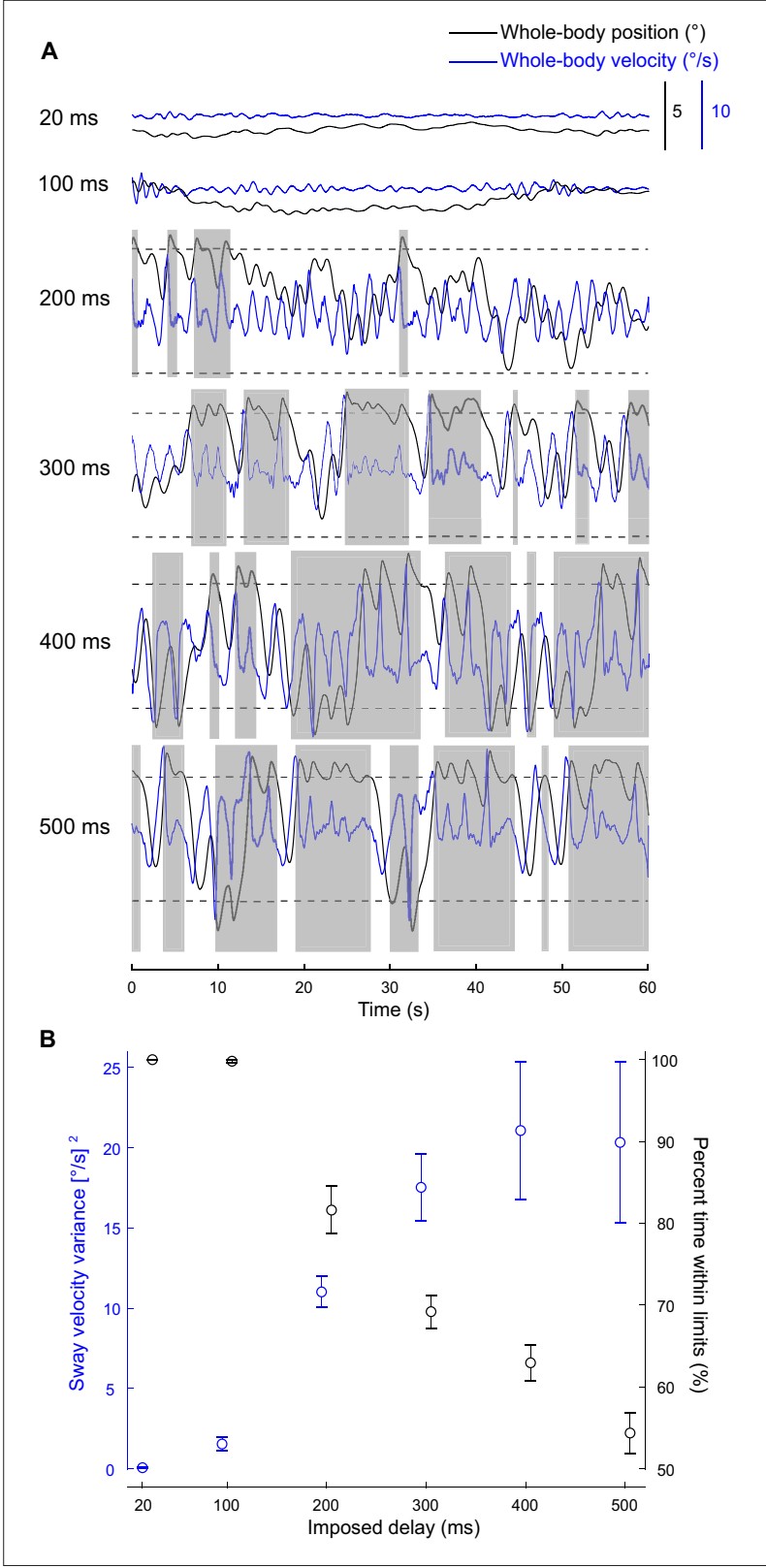

**Figure 2.** Standing balance behavior with delays. (**A**) Experiment 1: raw traces of body position (black) and velocity (blue) for a single participant balancing on the robotic simulator for 60 s at different imposed delay conditions. Dashed lines represent the virtual position limits (6° anterior, 3° posterior). Sway velocity variance was calculated over 2 s windows (extracted by taking segments when sway was within balance limits for at least two continuous

*Figure 2 continued on next page*

*Figure 2 continued*

seconds) and the resulting data were averaged to provide a single estimate per participant and delay (see Materials and methods). Data that are not grayed out represent periods where there is at least two continuous seconds of balance within the virtual position limits. The percentage of trial time participant's whole-body position remained within the limits was also quantified. (**B**) Group (n = 13) averages of sway velocity variance (blue) and percent time within balance limits (black). Error bars represent ± s.e.m.

to balance with the delay with sufficient training (i.e., >30 min) and this ability is partially retained 3 months later.

## Vestibular testing: sensorimotor delays decrease vestibular contributions to balance

During vestibular testing, we probed the vestibular contribution to soleus muscle activity by exposing participants (n = 8) to a non-painful electrical vestibular stimulus (EVS) while they balanced on the robot at different delays (20–500 ms; Materials and methods) before, after, and 3 months following training. Vestibular-evoked muscle responses are known to attenuate when actual sensory feedback does not align with expected estimates from balancing motor commands (*Héroux et al., 2015*; *Luu et al., 2012*). Therefore, we hypothesized that increasing the delay between ankle torques and body motion would progressively diminish the vestibular response. We further hypothesized that learning to control balance with imposed delays would allow the brain to update its sensorimotor estimates of balance motion and consequently increase the vestibular-evoked muscle responses. Frequency domain measures (coherence and gain; see Materials and Methods) were evaluated qualitatively using the pooled participant estimates because with delays ≥ 200 ms, single-participant coherence only exceeded significance at sporadic frequencies and significant coherence is needed to obtain a reliable gain estimate. Our time-domain measure (cross-covariance; see Materials and methods), which estimates the net vestibular contribution to muscle activity at all stimulated frequencies, was extracted on a participant-by-participant basis and used for statistical analysis. Participants exhibited the largest vestibular-evoked muscle responses (coherence, gain,and cross-covariance) for the 20 and 100 ms delay conditions, where significant coherence was observed at frequencies between 0 and 25 Hz and cross-covariance responses were characterized by short (~60 ms) and medium (~100 ms) latency peaks exceeding the 95 % confidence interval (see *Figure 4A*). Prior to learning, pooled coherence and gain decreased with imposed delays ≥ 200 ms, and coherence fell below the significance threshold at most

**Table 1.** Summary of statistical results.

| Variable | Delay F | p | Learning F | p | Delay × learning interaction F | p |
|---|---|---|---|---|---|---|
| *Sway velocity variance* | | | | | | |
| Exp 1: standing balance trials | $F_{(5,59.15)} = 14.98$ | < 0.001 | N/A | N/A | N/A | N/A |
| Exp 2: vestibular testing | $F_{(5,111.26)} = 33.89$ | < 0.001 | $F_{(2,113.19)} = 46.65$ | < 0.001 | $F_{(10,111.25)} = 5.72$ | < 0.001 |
| Exp 2: perceptual testing | $F_{(6,118.83)} = 31.00$ | < 0.001 | $F_{(2,121.47)} = 25.82$ | < 0.001 | $F_{(12,118.83)} = 2.08$ | = 0.023 |
| *Other variables* | | | | | | |
| Exp 1: percent within limits | $F_{(5,60)} = 127.48$ | < 0.001 | N/A | N/A | N/A | N/A |
| Exp 2: cross-covariance | $W_{(5)} = 1158.86$ | < 0.001 | $W_{(2)} = 70.57$ | < 0.001 | $W_{(7)} = 90.89$ | < 0.001 |
| Exp 2: perceptual threshold | N/A | N/A | $F_{(2,11.84)} = 7.52$ | = 0.008 | N/A | N/A |

For Exp 2, vestibular cross-covariance responses (peak-to-peak amplitudes) were analyzed using an ordinal logistic regression after rank transforming the data.

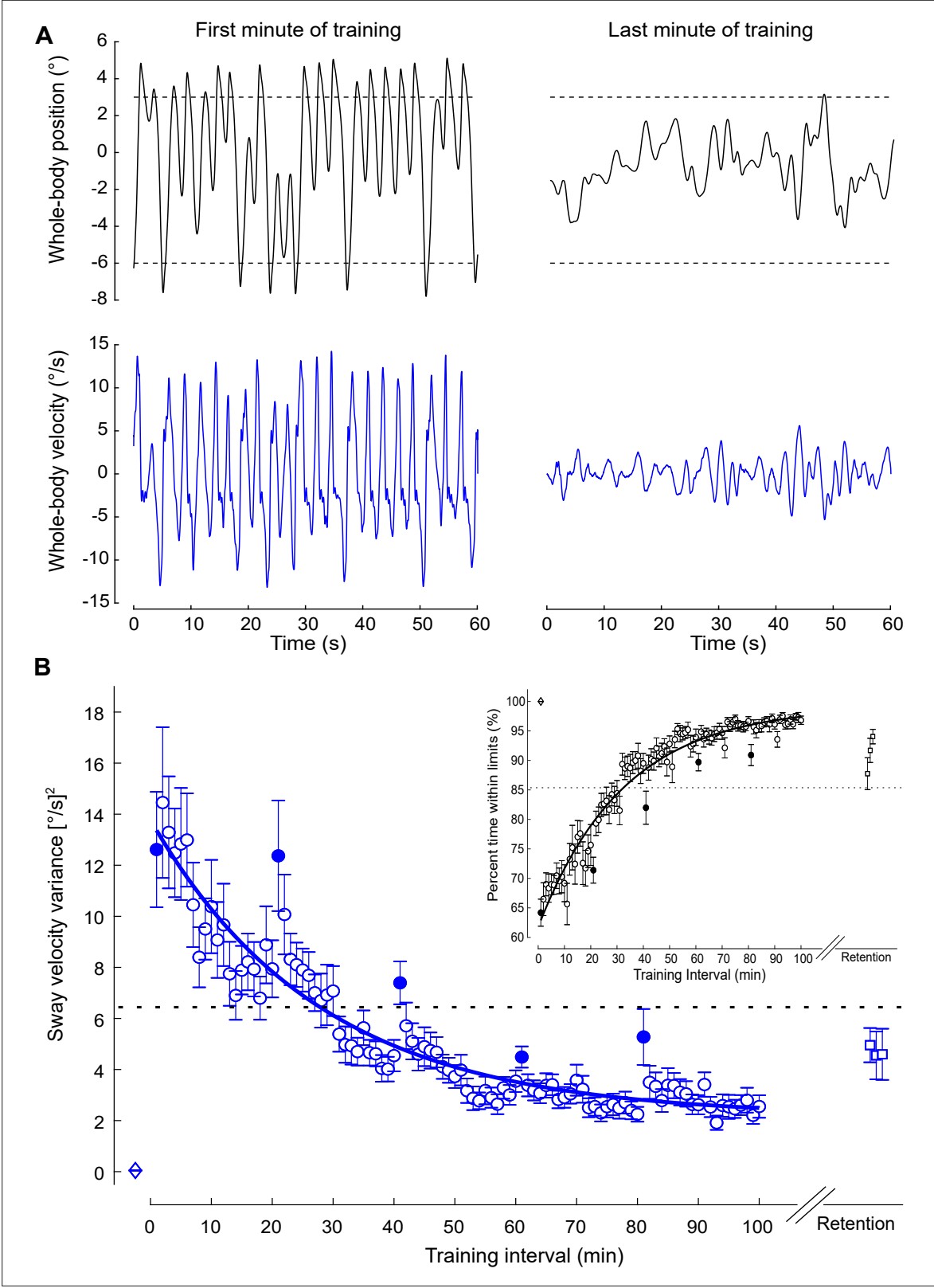

**Figure 3.** Standing balance behavior during the training protocol. (**A**) Whole-body position (°; black) and velocity (°/s; blue) traces of a representative participant when balancing in the first (left) and last (right) minute of training. During training trials, the robotic simulator operated with a 400 ms delay. (**B**) Average sway velocity variance and percentage of time spent within the virtual balance limits (inset) estimated over 1 min intervals during the 400 ms delay training (open circles) from all participants who completed the training protocol (n = 16). The first interval for each training session is represented

*Figure 3 continued on next page*

*Figure 3 continued*

by filled circles. Data from vestibular testing and perceptual testing groups were combined because both groups performed the same training protocol. Sway velocity variance progressively decreased and percentage of the interval time within the virtual limits progressively increased with each session of training (one session = 20 intervals). The solid lines show the fitting of sway velocity variance and percentage within virtual balance limits to a first-order exponential function using a least-square method: $f(x) = a * \exp\left(-\frac{x}{b}\right) + c$. For sway velocity variance, a = 11.61, b = 27.86, and c = 2.17; for percentage time within balance limits, a = –37.38, b = 32.45, and c = 99.12. The dashed horizontal lines represent the values at the estimated time constants. Data for the first minute of standing at 20 ms (open diamond) and the three minutes at 3 months after training (retention) at the 400 ms delay (open squares) are also presented. Error bars represent the s.e.m. for all data.

frequencies for delays ≥ 300 ms. Similarly, cross-covariance amplitudes decreased with increasing delay (≥200 ms), with only five out of eight participants showing significant biphasic muscle responses (cross-covariance) for the 400 ms delay condition (and six out of eight participants at 500 ms). Across training conditions (pre, post, retention), increasing the delay reduced the cross-covariance peak-to-peak amplitudes (main effect of delay, p<0.001; *Figure 4A and B*, *Table 1*).

Following training, pooled vestibular-evoked muscle responses (coherence, gain, cross-covariance) partially recovered in both the post-learning and retention phases. Every participant exhibited biphasic muscle responses that exceeded significance thresholds for every delay after training. A significant interaction between delay and learning was observed for the cross-covariance (p<0.001, *Table 1*), suggesting that the recovery of vestibular responses was dependent on the delay magnitude. Planned comparisons (Wilcoxon sign-rank test, Bonferroni corrected) revealed that cross-covariance response amplitudes were larger during post-learning relative to pre-learning for delays ≥ 200 ms (all p-values <0.05) and were larger during retention relative to pre-learning for 300 and 400 ms (p<0.05). Similar to Experiment 1, sway velocity variance generally increased with increasing delays (*Figure 4C*, *Table 1*, *Table 2*; p<0.001), while learning decreased sway velocity variance at almost all imposed delays (*Figure 4C*, *Table 1*; p<0.001), resulting in a significant delay × learning interaction (*Figure 4C*, *Table 1*; p<0.001). Planned comparisons (paired t-tests, Bonferroni corrected) revealed that sway velocity variance decreased during post-learning relative to pre-learning for delays between 20 and 400 ms (all p-values <0.05) and decreased during retention relative to pre-learning for delays between 100 and 400 ms (all p-values <0.05). Because training was only performed with the 400 ms delay, these training-related changes in vestibular responses (cross-covariance) and sway behavior across delays indicate that learning generalized to different sensorimotor delays.

## Perceptual testing: sensorimotor delays induce a perception of unexpected balance motion

During perceptual testing, we assessed whether participants (n = 18) perceived unexpected balance motion when transient delays (20–350 ms applied for 8 s periods; see Materials and methods) were imposed while balancing on the robot. Behavioral studies in humans suggest that delayed self-motion is perceived as unexpected (or externally imposed) because it does not align with prior expectations of the intended movement (*Blakemore et al., 1999*; *Farrer et al., 2008*; *Wen, 2019*). Therefore, we hypothesized that increasing the delay in the control of standing balance would evoke motion that is increasingly perceived as unexpected. We further hypothesized that learning to control balance with imposed delays would allow the brain to update its estimates of balance motion and consequently greater delays would be needed to elicit a perception of unexpected balance movements. When delays were transiently imposed during balance, whole-body sway became more variable and, as delays increased, participants perceived unexpected balance motion more often. Data from a representative participant (see *Figure 1E*) show missed detections of the 100 and 150 ms imposed delays, and this participant had a resulting 70 % correct detection threshold occurring at a delay of 136 ms. Across participants, the probability of perceiving unexpected postural motion increased with delays: from 4 % detection for the 50 ms delay up to 100 % for the 350 ms delay (*Table 3*). We found no significant difference in the 70 % correct detection thresholds for participants who did not participate in training (i.e., the no-learning group) and those who did during the pre-learning phase (156 ± 33 ms vs. 147 ± 21 ms; independent samples t-test: $t_{(16)}$ = 0.706, p=0.49). On average, when unexpected balance motion was correctly detected, participants pressed the button at least ~2 s after the delay was imposed across all delays and learning conditions (*Table 3*).

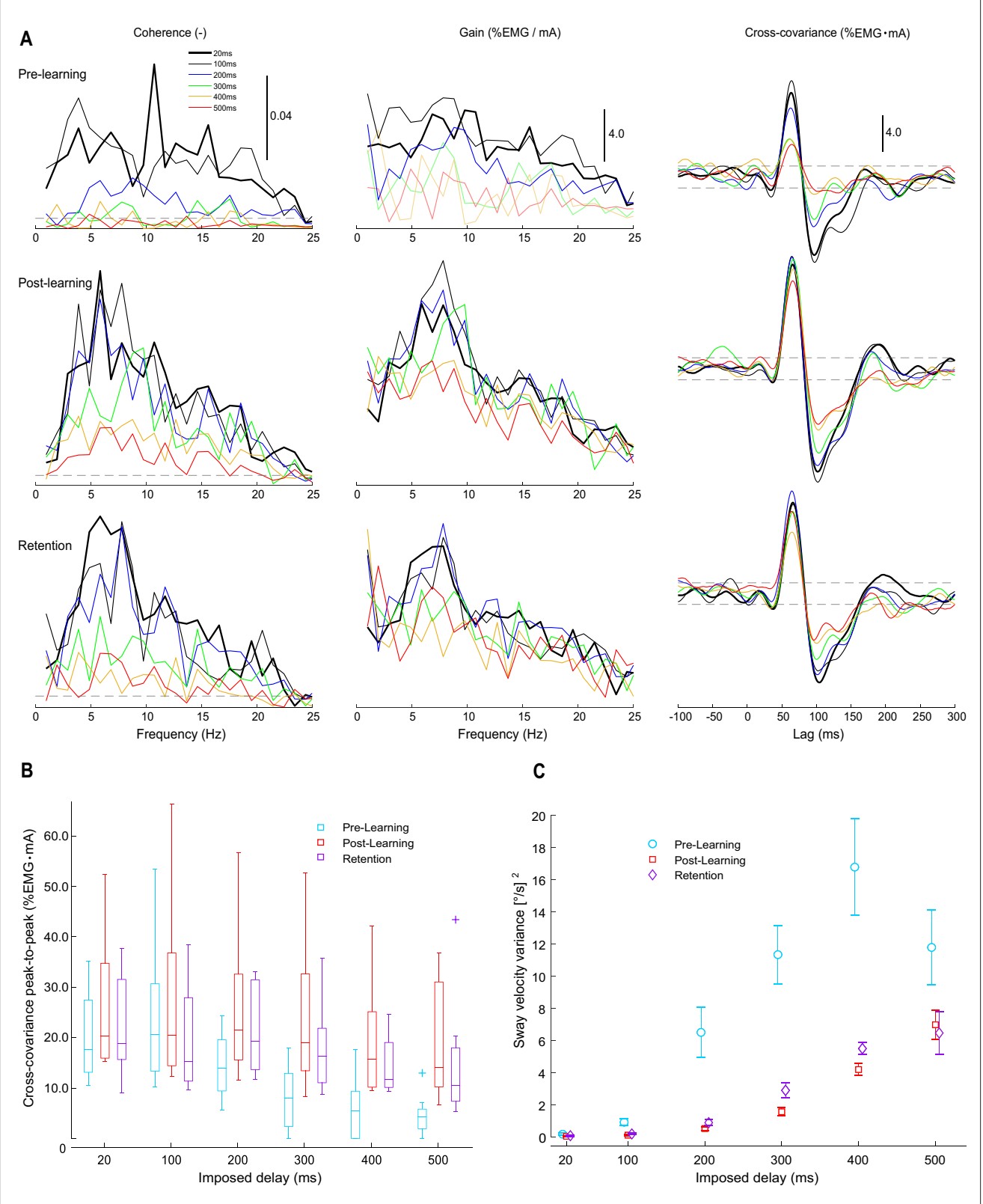

**Figure 4.** Experiment 2 vestibular-evoked muscle responses. Data are from pre-learning (n = 8), post-learning (n = 8), and retention (n = 7) conditions. (**A**) Coherence, gain, and cross-covariance between vestibular stimuli and rectified soleus EMG activity were calculated from the data concatenated from all participants. Estimates are presented from all six delay conditions (see legend). Horizontal dashed lines represent 95 % confidence limits for coherence and the 95 % confidence intervals for cross-covariance. Note that gain estimates are only reliable at frequencies with significant coherence;

*Figure 4 continued on next page*

*Figure 4 continued*

therefore, at delays ≥ 300 ms, where coherence falls below significance at most frequency points, the corresponding gain in the pre-learning condition was plotted using light lines. EMG was scaled by baseline EMG from each testing session (see Materials and methods), resulting in units for gain and cross-covariance of %EMG/mA and %EMG mA, respectively. (**B**) Group cross-covariance amplitudes (peak-to-peak) plotted relative to imposed delay. Across pooled estimates and group data, vestibular responses attenuated with increasing imposed delays and their amplitudes partially recovered after training. (**C**) Average sway velocity variance during vestibular stimulation trials. Non-normally distributed group data (vestibular response amplitudes) are plotted as medians (horizontal lines in boxes), 25 and 75 percentiles (boxes) and extreme data points (error bars). Normally distributed data (sway velocity variance) are presented as means with s.e.m (error bars).

After participants (n = 8) completed training with the 400 ms delay, psychometric functions shifted to the right, resulting in increased thresholds for detecting unexpected balance movements (*Figure 5A and B*, *Table 1*; p=0.008, Bonferroni corrected). Thresholds were larger during post-learning relative to pre-learning (*Figure 5B*; 192 ± 40 ms vs. 147 ± 21 ms; p=0.032, Bonferroni corrected) and were larger for retention relative to pre-learning (209 ± 82 ms vs. 147 ± 21 ms; p=0.014, Bonferroni corrected). For the whole-body oscillations (sway velocity variance extracted over the 8 s periods when delays were imposed, see *Table 1*), we again confirmed significant main effects of delay (p<0.001) and learning (p<0.001) as well as a delay × learning interaction (p=0.023). Planned comparisons (paired t-tests, Bonferroni corrected) revealed that sway velocity variance decreased during post-learning relative to pre-learning for delays ranging from 150 to 250 ms (all p-values <0.05) and decreased during retention relative to pre-learning for the 150, 200, 300, and 350 delays (all p-values <0.05).

## Experiment 3: rapid attenuation of vestibular responses accompanies balance variability and perceptual detection of unexpected balance motion

Our results from the vestibular and perceptual testing of Experiment 2 indicate that reduced vestibular responses (*Figure 4*) and a higher probability of perceiving unexpected balance behavior (*Figure 5*) were accompanied by increased sway velocity variance. These two experiments, however, involved different methodologies (trials with constant delays vs. transient changes in delay), making it difficult to compare sway behavior between data sets or determine if vestibular modulations coincided with perceptual changes. Therefore, in Experiment 3 we tracked the time course of vestibular responses and the occurrence of perceptual detections together with whole-body oscillations when delays were transiently imposed. Here, participants (n = 7) were exposed to a transient delay of 200 ms because Experiment 2 revealed that this delay increases sway variability, attenuates vestibular responses, and elicits frequent perceptions of unexpected standing motion. To quantify balance variability throughout

**Table 2.** Vestibular response magnitude and sway behavior from vestibular stimulation trials in Experiment 2 vestibular testing.

| Delay (ms) | 20 | 100 | 200 | 300 | 400 | 500 |
|---|---|---|---|---|---|---|
| *Pre-learning (n = 8)* | | | | | | |
| Cross-cov. (%EMG·mA) | 17.7/16.0 | 20.7/18.8 | 14.0/11.0 | 8.10/12.4 | 5.54/10.9 | 4.35/5.50 |
| Sway velocity variance [°/s]$^2$ | 0.18 ± 0.17 | 0.93 ± 0.58 | 6.52 ± 4.40 | 11.35 ± 5.10 | 16.79 ± 8.51 | 11.80 ± 6.53 |
| *Post-learning (n = 8)* | | | | | | |
| Cross-cov. (%EMG·mA) | 20.4/23.2 | 20.4/26.7 | 21.5/19.6 | 19.1/20.5 | 15.7/16.0 | 14.1/21.4 |
| Sway velocity variance [°/s]$^2$ | 0.05 ± 0.05 | 0.13 ± 0.09 | 0.54 ± 0.32 | 1.58 ± 0.74 | 4.20 ± 1.05 | 6.99 ± 2.61 |
| *Retention (n = 7)* | | | | | | |
| Cross-cov. (%EMG·mA) | 18.8/20.0 | 15.3/17.6 | 19.3/20.1 | 16.4/12.5 | 11.8/9.16 | 10.6/13.3 |
| Sway velocity variance [°/s]$^2$ | 0.09 ± 0.11 | 0.20 ± 0.12 | 0.90 ± 0.48 | 2.91 ± 1.25 | 5.51 ± 0.98 | 6.48 ± 3.51 |

Vestibular responses (peak-to-peak cross-covariance amplitudes) were not normally distributed and are presented as median/interquartile range.
Sway velocity variance were normally distributed and are presented as mean ± SD.

**Table 3.** Perceptual detection rates and sway behavior from perceptual testing in Experiment 2.

| Delay (ms) | 50 | 100 | 150 | 200 | 250 | 300 | 350 |
|---|---|---|---|---|---|---|---|
| *No learning (n = 10)** | | | | | | | |
| Used trials (out of 200) | 197 | 194 | 195 | 196 | 195 | 198 | N/A |
| Detections (% detected) | 8 (4%) | 60 (31%) | 128 (66%) | 172 (88%) | 186 (95%) | 198 (100%) | N/A |
| Sway velocity variance [°/s]$^2$ | 0.12 ± 0.05 | 0.57 ± 0.48 | 1.69 ± 1.21 | 3.71 ± 2.52 | 4.87 ± 2.62 | 6.32 ± 1.95 | N/A |
| Detection time (s) | 3.8 ± 2.0 | 4.7 ± 2.0 | 4.0 ± 1.9 | 3.5 ± 1.8 | 2.9 ± 1.5 | 2.6 ± 1.2 | N/A |
| *Pre-learning (n = 8)* | | | | | | | |
| Used trials (out of 160) | 148 | 151 | 147 | 147 | 150 | 151 | 152 |
| Detections (% detected) | 20 (14%) | 46 (30%) | 111 (76%) | 132 (90%) | 146 (97%) | 151 (100%) | 152 (100%) |
| Sway velocity variance [°/s]$^2$ | 0.24 ± 0.27 | 0.45 ± 0.33 | 1.84 ± 1.36 | 4.01 ± 2.33 | 4.18 ± 1.38 | 5.09 ± 1.46 | 4.70 ± 1.64 |
| Detection time (s) | 4.1 ± 2.1 | 3.7 ± 1.9 | 3.6 ± 1.8 | 3.2 ± 1.6 | 2.9 ± 1.6 | 2.4 ± 1.2 | 2.3 ± 1.1 |
| *Post-learning (n = 8)* | | | | | | | |
| Used trials (out of 160) | 157 | 156 | 157 | 156 | 157 | 157 | 151 |
| Detections (% detected) | 16 (10%) | 23 (15%) | 52 (33%) | 101 (65%) | 136 (87%) | 153 (97%) | 151 (100%) |
| Sway velocity variance [°/s]$^2$ | 0.11 ± 0.10 | 0.16 ± 0.13 | 0.40 ± 0.26 | 1.34 ± 1.09 | 2.20 ± 1.61 | 3.02 ± 2.33 | 3.70 ± 2.37 |
| Detection time (s) | 4.2 ± 2.0 | 3.8 ± 2.4 | 4.1 ± 1.7 | 3.9 ± 1.7 | 3.4 ± 1.8 | 2.7 ± 1.3 | 2.2 ± 1.1 |
| *Retention (n = 5)* | | | | | | | |
| Used trials (out of 100) | 96 | 93 | 98 | 98 | 96 | 98 | 92 |
| Detections (% detected) | 8 (8%) | 21 (23%) | 40 (41%) | 50 (51%) | 71 (74%) | 84 (86%) | 92 (100%) |
| Sway velocity variance [°/s]$^2$ | 0.05 ± 0.03 | 0.13 ± 0.06 | 0.27 ± 0.15 | 0.84 ± 0.51 | 1.51 ± 0.76 | 1.89 ± 1.09 | 2.60 ± 1.05 |
| Detection time (s) | 5.0 ± 1.7 | 4.2 ± 2.0 | 4.2 ± 2.2 | 3.8 ± 2.0 | 3.4 ± 1.8 | 3.1 ± 1.8 | 3.1 ± 1.4 |

Sway velocity variance and detection time are presented as mean ± SD.
*No learning group is an independent sample of participants that were not exposed to a 350 ms delay.

the transitions, sway velocity variance was estimated across a 2 s sliding window on a point-by-point basis (*Figure 6A*). Furthermore, to link changes in vestibular responses and perception, we compared vestibular response attenuation – estimated with a time-frequency analysis of coherence and gain (see Materials and methods) – at the time of perceptual detection of unexpected balance motion.

Out of the 588 total transitions, 489 were perceived as eliciting unexpected balance motion and used for all analyses. During periods of standing preceding the imposed delays, participants swayed with low-velocity variance (*Figure 6A*) and with consistent vestibular control of balance as shown by coherence and gain estimates between the vestibular stimulus and soleus EMG activity (*Figure 6B and C*). Transitions between baseline and delayed balance control increased whole-body sway velocity variance throughout most of the delay period (see *Figure 6A*). Over the same period, coherence and gain between EVS and EMG decreased. To characterize the time course of the decrease in this vestibular contribution to balance, we fit an exponential decay function to the mean coherence (i.e., coherence averaged over 0.5–25 Hz at each time point) from each participant. The 63.2 % attenuation (i.e.,

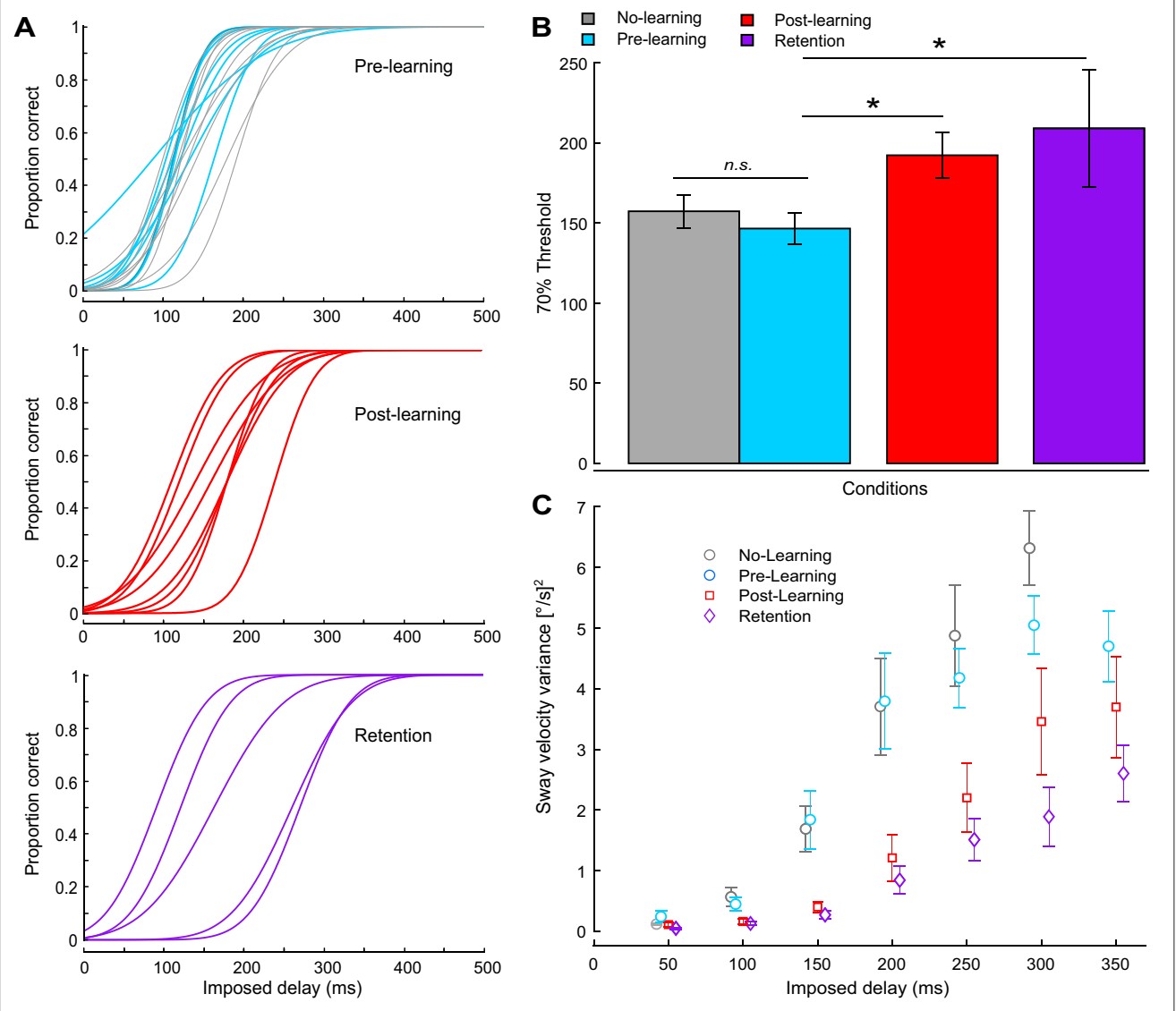

**Figure 5.** Experiment 2 perceptual testing and standing behavior results. (**A**) A Bayesian estimation procedure was used to fit sigmoidal functions to perceptual responses. The proportion of correct responses (i.e., button pressed during delay period) was calculated for each participant at each delay level. Individual psychometric functions are shown for all participants. The top panel shows participants tested before training (n = 18), with 10 participants who did not participate in the learning procedure shown in gray. The middle panel shows post-learning (n = 8), and the bottom panel shows retention results (n = 5). (**B**) Average 70 % interpolated threshold for pre, post, and retention conditions. Perceptual thresholds increased following training, such that larger imposed delays were needed to elicit perceptual detections. (**C**) Average velocity variance for different delays. Error bars indicate s.e.m.

the time constant) for coherence occurred at 1.5 ± 0.6 s following delay onset while the 95 % attenuation (i.e., 3× time constant) occurred at 4.4 ± 2.6 s following delay onset. For gain, we fit an exponential decay only to the mean gain estimated from the pooled data because for some participants coherence decreased below significance at all frequencies for some periods of the delay exposure (see *Figure 6C*, left-lower panel). The 63.2 % attenuation from this mean gain estimate occurred at 2.3 s while the 95 % attenuation occurred at 6.8 s. Perceptual detection times occurred over a similar time period, with the group averaged (489 detections) detection occurring at 3.4 ± 1.8 s after delay onset (see magenta line in *Figure 6*, right panel) and the 95th percentile for detection time occurring at 6.9 ± 0.8 s. Comparing the perceptual detection timing with the exponential decrease in coherence estimated from each participant, we found that the average time to detect the imposed delay (3.4 ± 1.8 s) corresponded to a 90% ± 7% average reduction in mean coherence. For the mean gain estimate

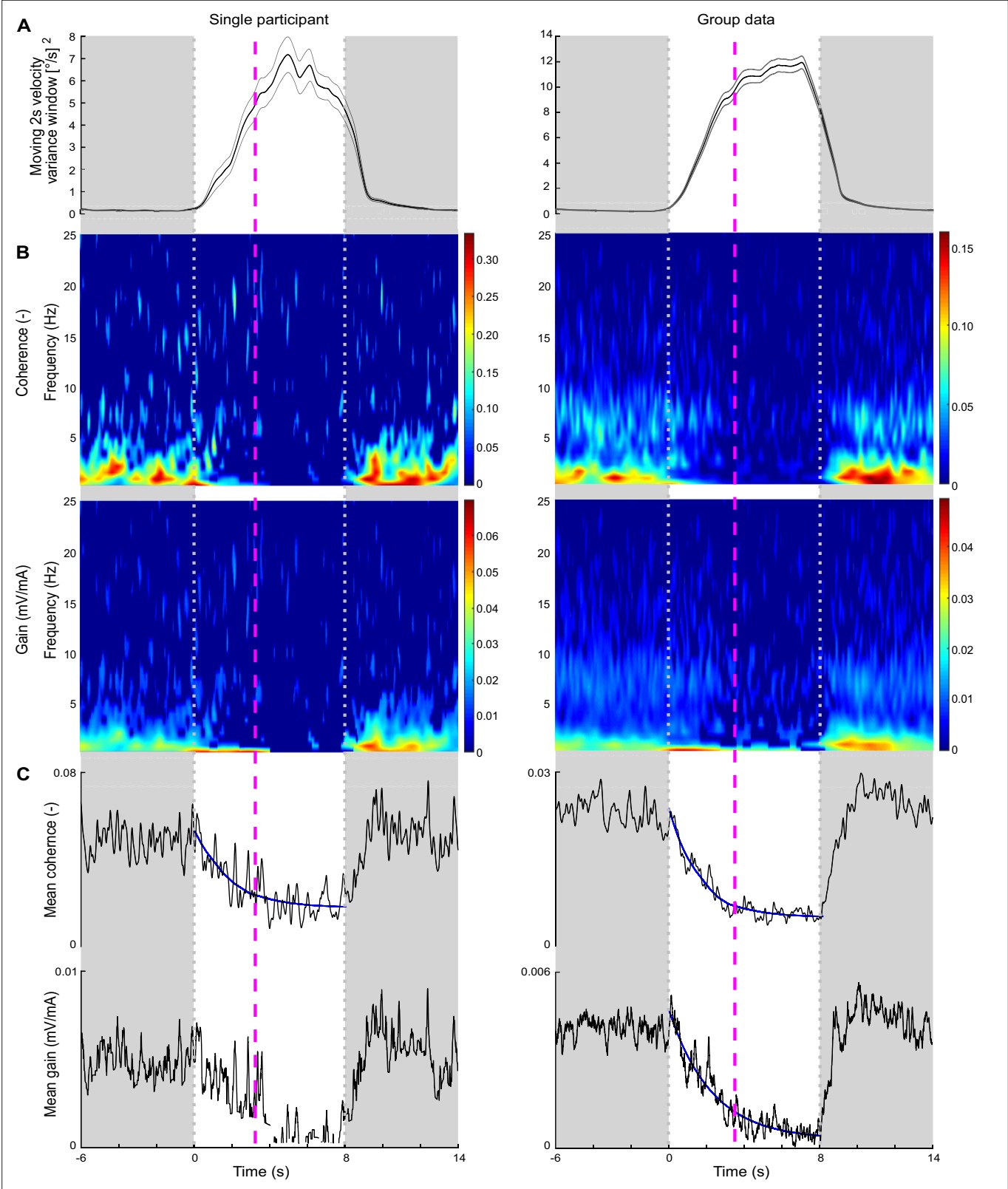

**Figure 6.** Experiment 3 sway velocity variance, time-varying electrical vestibular stimulus-electromyography (EVS-EMG) coherence and gain, and perceptual detection time during delay transitions. Data are presented across transition periods where time zero represents the transition point from baseline (20 ms) to 200 ms delayed balance control, which lasted for 8 s (between grayed out areas) and returned to baseline. Data are presented for a representative participant (left) and the group data (right; n = 7) using only data from transitions that were perceived as unexpected and the button was

*Figure 6 continued on next page*

*Figure 6 continued*

pressed after the delay was introduced (single participant: 77, group: 489). (**A**) Average (black line) 2 s sliding window of velocity variance over transitions with ±s.e.m. (gray lines). Time-varying variance was calculated using the *movvar* MATLAB function, which calculated variance over 2 s segments using a sliding window. The velocity variance trace begins to decline prior to the end of the delay period because the sliding window starts estimating variances from data points both during and after the delay. (**B**) Time-frequency plots of EVS-EMG coherence and gain (i.e., vestibular-evoked muscle responses) during the delay transition. For illustrative purposes, and because gain values are not reliable when coherence is below the significance threshold, we set coherence and gain data points where coherence was non-significant (i.e., below 99 % confidence limit) to zero (dark blue). (**C**) Mean time-dependent EVS-EMG coherence and gain across the 0.5–25 Hz frequency range. For each participant and group data, an exponential decay function: $f(x) = a * \exp\left(-\frac{x}{b}\right) + c$ was fit to the average coherence over the 8 s period during which the 200 ms delay was present. For gain, we removed values corresponding to non-significant coherence (see single participant trace) and only fit an exponential function to the group mean gain estimate. The average perceptual detection times for the representative participant (3.2 s) and group data (3.4 s) are indicated by the dashed magenta lines, and at these times, vestibulomuscular coherence had attenuated by 83% and 90%, respectively. Group mean gain attenuated by 73 % at the group average perceptual detection time.

(from all participants), this average perceptual time aligned with a 73 % reduction in gain. The peak velocity variance leading up to a perceptual detection was also extracted for each perceived transition and resulted in an average peak velocity variance of 8.42 ± 8.62 [°/s]². Overall, these results indicate that the perception of unexpected motion and increased sway variability arising from an imposed delay are accompanied by an ~70–90% attenuation of vestibular contributions to balance.

## Discussion

The primary aims of this study were to (1) characterize the destabilizing effects of imposed delays between ankle torque and whole-body motion during standing balance and (2) determine the underlying mechanisms responsible for adaptation and learning to these imposed delays. When delays were first imposed, the variance of whole-body sway velocity during balance increased with larger delays and all participants exceeded the virtual balance limits (6° anterior, 3° posterior) when delays were ≥200 ms. Balancing with imposed delays also attenuated vestibular-evoked muscle responses and led to perceptions of unexpected movement during standing balance, supporting the interpretation that imposing delays increased uncertainty in the internal estimate of balancing self-motion. Importantly, participants learned to balance with an imposed sensorimotor delay of 400 ms over 5 days (100 min of training), showing decreased sway velocity variance and increased percent time balancing within the virtual limits, partially restored vestibular control of balance, and fewer unexpected movement detections while balancing with the delay. These effects were generalized across delays despite training to balance only at the 400 ms delay. Our findings reveal that, while there may be a critical delay for the balance system, the brain can find a solution to overcome this limitation, learning to maintain standing balance with imposed delays by causally linking delayed whole-body sensory feedback that was initially interpreted as imposed motion to self-generated balance motor commands.

### Learning to stand with novel sensorimotor delays

When delays were first inserted in the control of standing balance, sway velocity variance increased with larger delays to the point that upright balance could not be maintained for at least 60 s at imposed delays ≥ 200 ms (i.e., total sensorimotor delay ≥300–360 ms). These results support predictions from computational models that upright standing is destabilized with added delays and that balance is impossible passed a critical delay of ~300–340 ms (*Milton and Insperger, 2019*; *van der Kooij and Peterka, 2011*). This critical delay was previously estimated by varying the parameters of a proportional-derivative (PD) feedback control model of standing balance, and suggests that even with training, the balance controller is incapable of adjusting the gain of its feedback to maintain upright stance with delays beyond this upper limit. Our results, in contrast, clearly demonstrate that participants improved their balance behavior (i.e., reduced sway velocity variance) when given the opportunity to train with a 400 ms delay (~63 % by 28–33 min), and that this improvement was retained, at least partially, 3 months later. Therefore, the nervous system can learn to control standing balance (although with more variability) with a net delay of up to 500–560 ms. The notable learning observed after training may have been partly due to participants being passively supported past the

virtual balance limits because it prevented certain nonlinear behaviors that disrupt continuous balance control such as taking steps or falls.

This remarkable ability for humans to adapt and maintain upright stance with delays raises questions regarding the principles underlying the neural control of balance. Compared to feedback controllers (i.e., PD and proportional-integral-derivative), which are not optimal in the presence of delays, optimal controllers can model the control of human standing (*Kiemel et al., 2002*; *Kuo, 1995*; *Kuo, 2005*; *van der Kooij et al., 1999*; *van der Kooij et al., 2001*) and theoretically stabilize human standing with large (>500 ms) delays (*Kuo, 1995*). This ability, however, rapidly declines with increasing center of mass accelerations (*Kuo, 1995*), including those driven by external disturbances (*Zhou and Wang, 2014*). Although feedback and optimal controllers assume that the nervous system linearly and continuously modulates the balancing torques to stand (*Fitzpatrick et al., 1996*; *Masani et al., 2006*; *van der Kooij and de Vlugt, 2007*; *Vette et al., 2007*), intermittent corrective balance actions (*Asai et al., 2009*; *Bottaro et al., 2005*; *Gawthrop et al., 2011*; *Loram et al., 2011*; *Loram et al., 2005*) may represent a solution when time delays rule out continuous control (*Gawthrop and Wang, 2006*; *Arsan et al., 1999*). Intermittent muscle activations are also sufficient to stabilize the upright body during a robotic standing balance task similar to the one used in the present study (*Huryn et al., 2014*). The nervous system may use a combination of the controllers during standing (*Elias et al., 2014*; *Insperger et al., 2015*), but our study did not explicitly test for evidence of these different controllers or their ability to stabilize upright stance with large delays.

## Learning to expect novel sway behavior caused by imposed sensorimotor delays

Increasing the imposed delay between balancing motor commands and the whole-body motion associated with those actions progressively attenuated vestibular-evoked muscle responses and led to more frequent perceptual detections of unexpected movement. The attenuation of vestibular-evoked responses to increased sensorimotor delays could be explained through processes of sensory reweighting (*Cenciarini and Peterka, 2006*; *Peterka, 2002*), where the decreasing reliability in sensory cues when balancing with additional delays decreases feedback gains. Indeed, feedback control models of standing predict that sensorimotor gains should decrease with increasing sensorimotor delays (*Bingham et al., 2011*; *Le Mouel and Brette, 2019*; *van der Kooij and Peterka, 2011*). However, our results show the partial return of vestibular response amplitude and shifted perceptual responses after training at a delay of 400 ms, indicating the involvement of alternative processes of sensorimotor recalibration. Both the vestibular and perceptual measures are influenced by whether the brain interprets sensory feedback as self-generated (expected) or externally imposed (unexpected), and their adaptation indicates that the nervous system learned to expect the delayed balancing-related feedback associated with self-generated balancing motor commands. Similar recalibration of the vestibular control of balance has been observed when humans stand with manipulated vestibular feedback gains (*Héroux et al., 2015*). When first standing with altered vestibular feedback (via a head-coupled EVS), variability in balance behavior increased and the amplitude of vestibular-evoked muscle responses (probed with an independent EVS signal) decreased. After a short (240 s) calibration period with the novel head-coupled vestibular stimulus, postural sway and vestibular responses returned to baseline levels (*Héroux et al., 2015*), suggesting that the brain learned to expect the modified vestibular feedback. Notably, this could not be explained by sensory reweighting mechanisms since adaptation did not occur when applying matching levels of EVS that were uncoupled from head motion. Perceptually, there is similar evidence of recalibration to new sensorimotor delays. Following repeated exposure to imposed sensorimotor delays, the perceived timing between motor actions and sensory feedback can be shifted according to the magnitude of the delay (*Stetson et al., 2006*) and the perceived intensity (i.e., force, tickle sensation) of delayed touch returns to baseline (i.e., no delay) levels (*Kilteni et al., 2019*). Our results suggest that these forms of recalibration are possible for the vestibular control of standing balance and the perception of standing balance. Because we naturally experience changing neural delays during growth and aging (*Eyre et al., 1991*), it is crucial that the nervous system adapts and recalibrates to unexpected temporal relationships between sensory and motor signals to maintain stable balance control and perception of standing movement.

What neural substrates could be responsible for the brain learning to expect standing balance feedback that is initially unexpected? When faced with a new sensorimotor relationship, deep cerebellar

neurons (rostral fastigial nucleus) initially increase their activity to vestibular signals because the motor commands result in unexpected sensory feedback (*Brooks et al., 2015*; *Brooks and Cullen, 2013*). Learning this new sensorimotor relationship results in a gradual increase in the probability that this unexpected feedback arises from desired motor commands, leading to a gradual return of the normal firing patterns of the deep cerebellar neurons (*Brooks et al., 2015*). These neuronal recordings further indicate that the nervous system scales responses (i.e., not a switch-like mechanism) to sensory inputs based on the relative probability that sensory feedback is caused by motor actions. Our vestibular experiment results reflect a similar mechanism – vestibular response amplitudes gradually decreased with the magnitude of the delay – extending this framework to a standing balance control context. Additionally, training led to modified vestibular and perceptual responses as well as decreased sway velocity variance that transferred to different imposed delays, indicating a generalized effect of learning. This suggests that the nervous system did not specifically recalibrate sensorimotor cues to the 400 ms delay but estimated the source of the unexpected balance motion (i.e., distorted motor command – whole-body motion relationship) and broadly updated its control (*Berniker and Kording, 2008*; *Braun et al., 2010*; *Krakauer and Mazzoni, 2011*) to accommodate for imposed delays.

## Rapid attenuation of vestibular responses accompanies perception and postural instability

Our third experiment showed that within a few seconds of balancing with a 200 ms delay, an ~70–90% attenuation of vestibular responses accompanied increased sway velocity variance and the perception of unexpected standing motion. Part of the variability in whole-body sway caused by sensory manipulations is considered to represent errors in balance estimates (*Kiemel et al., 2002*). Because imposed delays should evoke unexpected feedback errors (*Blakemore et al., 1999*; *Farrer et al., 2008*; *Haering and Kiesel, 2015*; *Wen, 2019*), the changes in vestibular-evoked muscle responses and perception with sway velocity variance may reflect that the two behavioral responses are linked to balance errors. However, it is not explicitly clear from our data what specific component of the standing behavior can be attributed to discrepancies between actual and expected feedback. Therefore, changes in standing behavior following adaptation to a delay could also be attributed to other factors induced by the imposed delays, such as a change in control policy or increased volitional control to balance (*Elias et al., 2014*; *Ozdemir et al., 2018*; *Peterson and Ferris, 2019*). When accounting for this limitation, it is also plausible that the observed changes in vestibular and perceptual responses were partially driven by whole-body motion (i.e., magnitude and variability) and not solely prediction errors. For instance, vestibular responses are known to be modulated by standing kinematics (*Day et al., 1997*; *Rasman et al., 2018*; *Son et al., 2008*) and perceiving balance motion is related to whether the experienced motion exceeds sensory detection thresholds (*Fitzpatrick and McCloskey, 1994*). The important questions regarding whether movement variability is attributed to control errors and their resulting influence on balance control, perception, and adaptation are inherently challenging for standing balance because the task is continuous and does not have an effective end-point target (thus no computable end-point error). Future studies using carefully designed sensorimotor manipulations of balance control (*Rasman et al., 2018*) with differing effects (perhaps directional) on unexpected errors and whole-body sway behavior may resolve these critical issues.

## Clinical implications

During aging and certain diseases (e.g., diabetic neuropathy or multiple sclerosis), the sensory and/or motor conduction times increase and may affect an individual's ability to maintain their body center of mass position within their base of support. According to our results, the standing balance system can be trained to accommodate a larger range of sensorimotor delays than previously predicted, which may be valuable for clinical populations who are thought to experience instability and an increased risk of falls as a result of increased neural delays. Adaptation to increased neural delays with aging, for example, is thought to occur through decreased sensorimotor gains and increased ankle stiffness (*Le Mouel and Brette, 2019*). Because our results show that training with an additional 400 ms delay can restore vestibular contributions of balance across a broad range of imposed delays (and are retained over a 3 month period), it may be possible to explore targeted rehabilitation of aging-related balance impairments by training with delays to restore and sustain sensory contributions to standing balance.

## Limitations and other considerations

We manipulated the delay between ankle-produced torques (measured from the force plate) and the resulting whole-body motion (angular rotation about the ankle joints). This manipulation altered the timing between the net output of self-generated balance motor commands (i.e., ankle torques) and resulting sensory cues (visual, vestibular, and somatosensory) encoding whole-body and ankle motion. However, the timing between motor commands and part of the somatosensory signals from muscles (muscle spindles and Golgi tendon organs) and/or skin (cutaneous receptors under the feet) that are sensitive to muscle force (and related ankle torque) or movements and pressure distribution under the feet were unaltered by the imposed delays to whole-body motion. This may have led to potential conflicts in the sensory coding of balance motion and may have influenced the ability to control and learn to stand with imposed delays. As methodologies to probe and manipulate the sensorimotor dynamics of standing improve, future experiments can be envisioned to replicate and modify specific aspects (i.e., specific sensory afferents) of the physiological code underlying standing balance. Such endeavors are needed to unravel the sensorimotor principles governing balance control.

## Conclusion

We observed that increasing the imposed sensorimotor delays in standing balance results in unstable standing balance, attenuated vestibular control of balance, and perceptual detections of unexpected motion. The nervous system, however, can adapt to novel sensorimotor delays and learn to maintain upright standing despite their initially disruptive influence. This learning is accompanied by vestibular contributions to balance partially returning and standing motion more likely to be perceived as self-generated. Thus, our results suggest that the nervous system can learn to control standing balance with added sensorimotor delays by causally relating delayed whole-body sensory feedback (initially deemed unexpected) with self-generated balancing motor commands.

# Materials and methods

## Participants

A total of 46 healthy adult participants (32 males, age: 24.0 ± 3.9 years [mean ± SD]; range 19–34 years) with no known history of neurological deficits participated in this study. The experimental protocol was verbally explained before the experiment and written informed consent was obtained. The experiments were approved by the University of British Columbia Human Research Ethics Committee and conformed to the Declaration of Helsinki, with the exception of registration to a database.

## Experimental setup

Three experiments were conducted to investigate how imposed sensorimotor delays affect standing balance. We first performed an experiment evaluating standing balance behavior when participants balanced upright with different imposed sensorimotor delays. We then conducted a second experiment to determine (1) whether participants could learn to maintain standing balance with a 400 ms delay and (2) whether learning to control balance would lead to changes in the vestibular control of balance (Experiment 2 – vestibular testing) and perception of postural behavior (Experiment 2 – perceptual testing). To probe the vestibular control of balance, we delivered EVS while participants balanced upright with different delays. To assess perception, we applied transient delays during ongoing standing balance and asked participants to report when they consciously perceived unexpected postural motion. Finally, we performed a third experiment to track the time course of modulations in vestibular contributions to balance caused by imposed delays and how it follows changes in sway behavior variability and perception of unexpected balance motion.

For all experiments, participants stood on a custom-designed robotic balance simulator programmed with the mechanics of an inverted pendulum to replicate the load of the body during standing (*Figure 1A*). Specifically, the simulator used a continuous transfer function that was converted to a discrete-time equivalent for real-time implementation using the zero-order hold method

$$I\ddot{\theta} - m_m gL\theta = T$$

$$\frac{\theta}{T} = \frac{1}{Is^2 - 0.971mgL}$$

as described by *Luu et al., 2011*, where $\theta$ is the angular position of the body's center of mass relative to the ankle joint from vertical and is positive for a plantar-flexed ankle position, $T$ is the ankle torque applied to the body, $m_m$ is the participant's effective moving mass, $L$ is the distance from the body's center of mass to the ankle joint, $g$ is gravitational acceleration (9.81 m/s$^2$), and $I$ is mass moment of inertia of the body measured about the ankles ($m_mL^2$). The body weight above the ankles was simulated by removing the approximate weight of the feet from the participant's total body weight so that the effective mass was calculated as $0.971m$, where m is the participant's total mass. The balance simulator was controlled by a real-time system (PXI-8119; National Instruments, TX, USA) running at 2000 Hz and consisted of an ankle-tilt platform and rigid backboard independently controlled by two rotary motors (resolution of 0.00034°; SCMCS-2ZN3A-YA21, Yaskawa, Japan). The backboard was lined with a layer of medium-density foam and memory foam. Participants were secured to the backboard through seat belts at the waist and shoulders, and the backboard orientation was adjusted relative to the frame to account for the participant's natural standing posture. Participants stood on a force plate (OR6-7-1000; AMTI, MA, USA) secured to the ankle-tilt platform. In all experiments, the ankle-tilt platform was held horizontal (earth-fixed reference) while the backboard moved the upright body in the AP direction in response to ankle plantar- and dorsiflexion torques, thus replicating whole-body AP movements associated with standing (*Luu et al., 2011*). The delay between a position command and the measured position of the motor was estimated to be 20 ms (*Shepherd, 2014*). A visual projection screen was located to the left of the robotic device on which a 3D scene of a city courtyard with a water fountain was presented (Vizard 2013 software; WorldViz, CA, USA). Rendering and projection of the visual scene took approximately 70 ms; therefore, a linear least-squares predictor algorithm was used to synchronize the visual motion (i.e., predict visual motion occurring 50 ms later) together with the motors at a delay of 20 ms (*Shepherd, 2014*). The linear prediction model used six data points, and the coefficients were selected by fitting data of participant sway to the corresponding data shifted by the appropriate delay using a linear least-squares method. Participants wore active 3D glasses (DLP Link 3D Glasses; BenQ, Taipei, Taiwan), modified to block out peripheral vision and limit the participant's field of view to approximately ±45° horizontally and ±30° vertically. Participants also wore earplugs and noise canceling headphones (Bose Soundlink Around, Bose Corporation, MA, USA) with audio of a water fountain to minimize acoustic cues of motion produced by the motors as well as other extraneous sounds. To represent the physical limits of sway during standing balance, the backboard rotated in the AP direction about the participant's ankles with virtual angular position limits of 6° anterior and 3° posterior (*Luu et al., 2011*; *Shepherd, 2014*). When the backboard position exceeded these position limits, the program gradually increased the simulated stiffness such that the participants could not rotate further in that direction regardless of the ankle torques they produced. This was implemented by linearly increasing a passive supportive torque to a threshold equivalent to the participant's body load over a rotation range of 1° beyond the balance limits (i.e., passively maintaining the body at that angle). Any active torque applied by participants in the opposite direction would enable them to get out of the limits. Finally, to avoid a hard stop at these secondary limits (i.e., 7° anterior and 4° posterior), the supportive torque was decreased according to an additional damping term that was chosen to ensure a smooth attenuation of motion.

The balance simulation was modified to add a delay between the measured ankle torque (i.e., motor command) and whole-body sway (i.e., sensory feedback). Delays were programmed by buffering ankle torque recordings such that the signals driving motor position commands, and therefore whole-body sway, could be delivered based on the torque participants performed up to 500 ms in the past. The natural sensorimotor delays within the standing balance controller are estimated to be ~100–160 ms (*Forbes et al., 2018*; *Kuo, 2005*; *van der Kooij et al., 1999*). Here, as our delays are in the robotic system, they need to be added to the internal delays to estimate the overall standing balance delays. Throughout this study, we refer to the delays added to the robotic simulator (20–500 ms), but the total sensorimotor delays for the standing balance task are ~100–160 ms larger. All participants were naïve to the delay protocols and were simply told that 'at times during the balance simulation, control may change such that your body movement may seem unexpected or abnormal, and standing balance may become more difficult.' In all experiments, participants were instructed to stand upright normally at their preferred standing angle (typically ~1–2° anterior). In trials with delays ≥ 200 ms, participants had difficulty maintaining a stable upright posture and would often reach the virtual balancing limits (i.e., 6° anterior or 3° posterior). When this happened, participants were immediately instructed by the

experimenter to get out of the limits and continue to balance upright. Despite these efforts, trials with larger imposed delays were often characterized by brief periods (~2–5 s) of active balancing before crossing the virtual balancing limits (i.e., angular whole-body position exceeding the angular position limits). After a trial was completed, the robot was returned to a neutral position (0°) at a fixed velocity (0.5°/s) in preparation for the next trial.

## Data recordings

Signals used to control the robotic simulation in real time were processed via a multifunction reconfigurable module (PXI-7853R; National Instruments, Austin, TX, USA). The backboard motor encoder provided backboard angular position data regarding the participant's whole-body position during the task. The delay (ms) used during the simulation was recorded for all experiments. Force plate signals (forces and torques) were amplified ×4000 (MSA-6; AMTI, Watertown, MA, USA) prior to being digitized. Vestibular stimuli (see Experiment 2 – vestibular testing and Experiment 3) and button switch signals (see Experiment 2 – perceptual testing and Experiment 3) were also recorded. For Experiment 2 – vestibular testing and Experiment 3, surface electromyography (EMG) was collected from the right soleus to measure the vestibular-evoked responses. We measured activity from this muscle for two reasons: (1) the head was turned left during all experiments and this orientation aligns the vestibular-evoked error with soleus muscle's line of action (see Vestibular stimulation) and (2) the soleus was shown to have the most consistent activity during pilot testing at the manipulated delay conditions, which is a prerequisite to estimate a vestibular-evoked response in lower-limb muscles (*Dakin et al., 2007*; *Forbes et al., 2013*). The skin over the muscle was cleaned with an alcohol swab and abraded with gel (Nu-Prep, Weaver and Company, Aurora, CO, USA). Self-adhesive Ag-AgCl surface electrodes (Blue Sensor M, Ambu A/S, Ballerup, Denmark) were positioned over the belly of the muscle in a bipolar configuration, with an inter-electrode distance of 2 cm center-to-center. For each participant, we noted the electrode placement on the soleus (by measuring the distance from the electrodes to the heel) during the first experimental session and used the same placement for consistency across experimental sessions (pre-learning, post-learning, retention). To reduce electrical noise from the motors, two ground electrodes were used: a nickel-plated disc electrode coated with electrode gel (Spectra 360 Parker Laboratories, NJ, USA) secured to the right medial malleolus, and a Velcro strap electrode secured around the right lower leg overlaying the tibial tuberosity and head of the fibula. Surface EMG signals were amplified (×5000, Neurolog, Digitimer Ltd., Hertfordshire, UK) and band-pass filtered (10–1000 Hz) prior to digitization. Force plate, vestibular stimuli, EMG, and button switch signals were recorded via a data acquisition board (PXI-6289; National Instruments). All signals were digitized at 2000 Hz.

## Familiarization

For all experiments, a balance session was first completed to familiarize the participant with the control of the robot. Instructions were given on the nature of movement control; that is, similar to standing, applying torque to the support surface (force plate) will control the motion of the upright body (backboard). In a forward leaning position, plantar-flexor torque is required to stabilize the body and an increase in plantar-flexor torque greater than the gravitational torque will cause the body to accelerate backward. Similarly, a dorsiflexor torque is required when standing in a backward leaning position and an increase in dorsiflexor torque will accelerate the body forward. Participants were instructed to sway back and forth and allow the robot to reach its limits (6° anterior, 3° posterior), which occurs if the magnitude of the generated ankle torque is not large enough to resist the toppling torque of gravity. After becoming familiar with the control of the robot, participants were then asked to stand quietly and maintain an upright posture (normal standing). Participants were also instructed to focus on the sensation of balance control and told that this setting was the 'normal' condition, which was of particular importance for perception experiments (Experiment 2 – perceptual testing and Experiment 3). Participants performed this familiarization period until they were accustomed to the sensation of standing balance on the robot. This ensured that participants could confidently discern between normal and novel (i.e., unexpected) standing balance behavior caused by manipulating delays. They completed the entire familiarization session within 5–7 min.

## Vestibular stimulation

In Experiment 2 – vestibular testing and Experiment 3, we used transmastoid EVS to probe vestibular-evoked muscle responses. Vestibular stimulation was delivered in a binaural bipolar configuration to modulate the firing rates of all vestibular afferents (*Goldberg et al., 1984*; *Kim and Curthoys, 2004*; *Kwan et al., 2019*) and provide an isolated vestibular error signal that evoked a virtual sensation of head motion about an axis-oriented posteriorly and superiorly by ~17–19° above Reid's plane (*Fitzpatrick and Day, 2004*; *Peters et al., 2015*). The head was pitched up such that Reid's plane was oriented ~17–19° up from horizontal. In this head position, EVS evokes a net signal of angular head rotation orthogonal to gravity (*Chen et al., 2020*; *Fitzpatrick and Day, 2004*; *Schneider et al., 2002*) and, through integration with an internal estimate of gravity, an inferred interaural linear acceleration signal (*Khosravi-Hashemi et al., 2019*). While standing, this imposed vestibular error evokes stereotypical compensatory muscle and whole-body responses (*Dakin et al., 2007*; *Day et al., 1997*; *Fitzpatrick and Day, 2004*; *Forbes et al., 2014*). The head was also turned ~90° to the left, which aligns the vestibular-evoked error signal with the AP direction of balance and line of action for the soleus, maximizing the muscle response (*Dakin et al., 2007*; *Forbes et al., 2016*). We chose stochastic vestibular stimuli (a white noise signal low-pass filtered to contain a set bandwidth of frequencies) rather than square-wave stimuli as it improves signal-to-noise ratio and reduces testing time (*Dakin et al., 2007*; *Reynolds, 2011*). EVS signals with a 0–25 Hz frequency bandwidth were generated offline using custom-designed computer code (LabVIEW 2013, National Instruments). The stimuli were delivered to participants through carbon rubber electrodes (9 cm$^2$) coated with Spectra 360 electrode gel and secured over the mastoid processes in a binaural bipolar configuration. The stimuli were sent as analog signals via a data acquisition board (PXI-6289, National Instruments) to an isolated constant current stimulator (DS5, Digitimer, Hertfordshire, England). Throughout the trials, head position was monitored and maintained using a head-mounted laser and verbal feedback, respectively.

## Experiment 1

Computational models indicate that adding sensorimotor delays will destabilize balance control (*Insperger et al., 2015*; *Milton and Insperger, 2019*; *van der Kooij et al., 1999*) and that upright standing cannot be maintained past a critical delay of ~340 ms (*Milton and Insperger, 2019*; *van der Kooij and Peterka, 2011*). To determine how imposed balance delays destabilize and limit balance control, 13 participants balanced the robotic system for 60 s with different imposed delays (20, 100, 200, 300, 400, and 500 ms). Trial order was the same for each participant, with delays presented in ascending order to avoid crossover effects from larger delays to smaller delays. Participants were not given any information regarding the different delay conditions or strategies on how to best control the robot.

## Experiment 2

Based on observations from Experiment 1, we designed a training protocol (Experiment 2) to determine if participants could learn to stand when the robotic balance control simulation operated with 400 ms delay (i.e., a total feedback delay of 500–560 ms). This delay was chosen because it is larger than the ~300–340 ms critical feedback delay of standing balance proposed previously (*Milton and Insperger, 2019*; *van der Kooij and Peterka, 2011*). To investigate the underlying physiological mechanisms of this learning process, we conducted Experiment 2 to evaluate how imposed delays influenced the vestibular-evoked balance responses and perception of standing motion, respectively, before and after training as well as 3 months later.

## Training procedure and timeline

Sixteen participants (vestibular testing, n = 8; perceptual testing, n = 8) performed the training protocol over five consecutive days (see *Figure 1C*). On the first day, prior to any training, participants completed pre-learning session (vestibular or perceptual testing; see below). Each training session then started with a 60 s standing balance trial with the 20 ms delay. This was followed by two 10 min training trials at the 400 ms delay. To minimize fatigue, participants rested for 2–3 min between these two trials. Participants were not given any specific instruction on how to improve their balance. On each day, the training trials were followed by a final 60 s trial at the 20 ms delay. In total, participants performed 100 min of training over 5 days at the 400 ms delay condition. After finishing the training

session on the fifth day, participants completed the post-learning session (vestibular or perceptual testing). Finally, 12 of the 16 participants (seven vestibular, five perceptual) returned ~3 months later (range: 81–110 days) to perform a retention session. The retention testing session began with a short familiarization period (<60 s) of balancing the robot with the 20 ms delay. Participants then completed vestibular or perceptual testing followed by two 3 min standing balance trials: one with the 20 ms delay and one with the 400 ms delay (order randomized between participants).

## Vestibular testing

Eight of the 16 training participants in Experiment 2 completed vestibular testing (*Figure 1C*). Here, we probed the vestibular-evoked responses in the soleus muscle while participants maintained standing balance at six imposed delays (see below). Vestibular testing was conducted for pre-learning, post-learning, and retention sessions. Seven participants returned ~3 months after the post-learning session to complete the retention session. Because vestibular-evoked responses are attenuated when actual sensory feedback does not align with estimates from balancing motor commands (*Héroux et al., 2015*; *Luu et al., 2012*), we hypothesized that increasing the delay between ankle torques and body motion would progressively attenuate the vestibular response. We further hypothesized that learning to control balance with imposed delays would allow the brain to update its sensorimotor estimates of balance motion and consequently increase the vestibular-evoked muscle responses.

For all vestibular response testing, participants stood on the robotic balance simulator with different delays while being exposed to EVS (*Figure 1D*). Six delay conditions were tested: 20, 100, 200, 300, 400, and 500 ms. Each trial began with a short period of data collection (~3–5 s) while the participants stood quietly on top of the robotic balance simulator, after which the electrical stimulus was delivered. For each delay condition, EVS was delivered in four 20 s trials, resulting in a total of 80 s of data (24 total trials). We performed short trials to minimize potential adaptation during a trial. Four different EVS signals (0–25 Hz bandwidth) were generated offline with root-mean-square (RMS) ranging from 1.47 to 1.61 mA (due to stochastic variation in signal generation) and each EVS stimulus was delivered once per delay condition (24 trials). We presented the trials in four subgroups, each containing the six delay conditions ordered randomly.

## Perceptual testing

A total of 18 participants, 8 of which participated in the training protocol of Experiment 2, completed perceptual testing (*Figure 1C*). Here, we examined if participants perceived unexpected postural behavior while being exposed to intervals of imposed sensorimotor delays (see below). All participants completed the pre-learning session, eight then completed the training protocol and were tested in the post-learning session and five of those eight returned ~3 months later for the retention session. If adding balance delays reduces the probability that sensory feedback is associated with motor commands, we hypothesized that participants would perceive self-generated balance oscillations as unexpected motion under delayed balance conditions. If training to stand with the imposed delay allows the brain to update its sensorimotor estimates of balance motion, we hypothesized that after training participants would be less prone to detecting unexpected standing motion when balancing with experimental delays (i.e., psychometric functions would shift rightward).

During the perception trials, the delay imposed by the robotic simulator transitioned while participants were actively balancing (see *Figure 1E*). Participants were instructed to indicate by pressing and holding a hand-held button switch when they perceived unexpected balance movements and release the button when balance felt normal. Delays were manipulated using a variation of the method of constant stimuli. We used seven different delays in the trials. For the 10 participants who did not participate in the training protocol (no-learning group), the delays were 20 (catch trial), 50, 100, 150, 200, 250, and 300 ms. We limited the delay to a maximum of 300 ms because preliminary testing showed that participants always perceived unexpected balance at the 300 ms delay. Participants that partook in the training protocol were presented with 50, 100, 150, 200, 250, 300, and 350 ms delays. We expected that after training some 300 ms delay periods would not elicit perceptions of unexpected motion (see Results), and therefore increased the experimental delay to a maximum of 350 ms. Participants balanced themselves in the robotic simulator for ~260 s in 10 separate trials. While participants actively balanced, we randomly inserted delays in the balance control for 8 s. During each trial, the robot transitioned instantaneously from the baseline 20 ms balance delay to one of the experimental

delays (20–350 ms) before returning to 20 ms. The inter-transition interval varied randomly between 7 and 10 s. In this manner, 14 delay periods (two of each delay level) were presented in a random order for each trial, resulting in a total of 20 delay periods for each experimental delay (yielding a 5 % resolution). The catch delay period (20 ms) from the no-learning group did not reveal any different performance from the existing inter-delay intervals and both the no-learning and pre-learning participants had 100 % success at detecting the 300 ms delay. Therefore, we present the psychometric functions (see *Psychometric functions*) of these two groups together.

## Experiment 3

Our results from the vestibular and perceptual testing in Experiment 2 showed that with added delays, (1) vestibular-evoked muscle responses are attenuated and participants perceive unexpected behavior, and (2) both vestibular and perceptual response modulations occur together with increases in sway variability (i.e., sway velocity variance) (see Results). Vestibular stimulation trials in Experiment 2 – vestibular testing had single, fixed delays (no transition between delays within the trial); consequently, we could not assess the time course of vestibular response attenuation. In Experiment 3, we tracked the time course of the vestibular, balance, and perceptual responses to imposed delays by repeatedly exposing participants (n = 7) to transient periods (8 s) of a 200 ms delay period while delivering EVS and instructing them to report perceptions of unexpected balance. Specifically, we determined (1) vestibular response attenuation in relation to when participants perceive unexpected standing behavior, and (2) the sway velocity variance associated with modulations in the vestibular and perceptual responses. Participants balanced on the robotic balance system for six separate trials lasting ~260 s each and were instructed to indicate perceived changes in balance control via button press. To probe the vestibular-evoked muscle responses, right soleus EMG was recorded and stochastic EVS was delivered (RMS: 1.38 mA) throughout each trial. The trials were similar to those of Experiment 2 – perceptual testing, except that only the 200 ms experimental delay was presented. We chose 200 ms based on our results from Experiments 1 and 2 (see Results). This specific delay increases sway velocity variance, attenuates vestibular-evoked balance responses, and elicits frequent perceptual detections (~80%) of unexpected standing motion. During each trial, the robot transitioned from the baseline balance condition (20 ms delay) to a 200 ms delay before returning to baseline. Transition to the delayed period occurred randomly over an inter-transition interval of 8–9 s and each delay period lasted exactly 8 s. Each trial consisted of 14 delay periods and a total of six trials were performed, providing 84 delay periods.

## Data reduction and signal analysis

All non-statistical processing and analyses were performed using custom-designed routines in MATLAB (2018b version, MathWorks, Natick, MA, USA) and LabVIEW software (LabVIEW 2013, National Instruments).

## Balance behavior

To quantify how balance behavior was affected by imposed delays in Experiments 1 and 2, we estimated the variance of whole-body sway velocity within each trial. Sway velocity variance was estimated only from data in which whole-body angular position was within the virtual balance limits (6° anterior and 3° posterior). Specifically, data were extracted in non-overlapping 2 s windows, when there was at least one period of two continuous seconds of balance within the virtual limits. Data windows were only extracted in multiples of 2 s. For instance, if there was an 11 s segment of continuous balance, we only extracted five 2 s windows (i.e., first 10 s of the segment). Sway velocity variance was estimated over these 2 s windows (see *Figure 2A*) because (1) participants could only balance within the balance limits for periods of ~2–5 s during some trials with delays ≥ 200 ms and (2) the transient delays in perceptual testing lasted only 8 s. Although too short to evaluate low-frequency postural sway position, this analysis window was considered appropriate to evaluate the variance of sway velocity because the velocity signal primarily consists of frequencies > 0.5 Hz (*van der Kooij et al., 2011*). On a participant-by-participant basis, we then averaged the sway velocity variance from the 2 s windows to provide an estimate of sway velocity variance for each participant in each balance condition (e.g., Experiment 1, 200 ms delay). For Experiments 1 and 2 (training trials), we also computed the percentage of time (over 60 s intervals) that whole-body sway position was within the

virtual balance limits. For the training trials in Experiment 2, sway velocity variance was estimated from non-overlapping 2 s windows taken across 1 min intervals (thus a maximum of 30 available windows per minute) throughout the training and retention phases, as well as during standing balance prior to training. These sway velocity variances were then averaged for every minute across all participants, providing a minute-by-minute representation of sway velocity variance in the training trials. For perceptual testing, where delays transitioned during perception trials, we evaluated the variance of sway velocity from 2 s windows over the period during which a delay was imposed. Since each delay was presented 20 times, this provided 160 s of data for each delay.

For Experiment 3, we identified peak sway velocity variance leading up to a perceptual detection. We therefore used a 2 s sliding window to extract the time course of sway velocity variance. Time-varying variance was calculated using the *movvar* MATLAB function, which calculated variance over 2 s segments while repeatedly moving over the data on a point-by-point basis. On a participant-by-participant basis, all delay periods (84 total) were classified as detected (button pressed during delay) or missed (button not pressed during delay). Using only perceptually detected trials (499 out of 588), we then extracted peak sway velocity variance over a period starting at the onset of the delay and ending at the perceptual detection from each perceived trial. Transitions where a button press occurred before delay onset were removed from the analysis (group data: 10 out of 499 transitions), providing 489 transitions in total.

## Vestibular-evoked muscle responses

To estimate the presence and magnitude of vestibular-evoked muscle responses in Experiment 2 - vestibular testing, we used a Fourier analysis to estimate the relationship between vestibular stimuli and muscle activity in the frequency and time domains. Specifically, we computed the coherence, gain, and cross-covariance functions between the electrical stimulus and soleus EMG for each participant (*Dakin et al., 2007*). Because vestibular stimulation trials were performed on three separate days (pre-learning, post-learning, retention sessions), we scaled soleus EMG to each session's baseline EMG measure. Specifically, for each testing session (e.g., pre-learning), we estimated the mean soleus EMG amplitude from a baseline quiet standing trial. We calculated the mean EMG amplitude recorded while participants maintained standing balance within ±0.25° around their preferred standing posture. We then scaled soleus EMG recorded during the vestibular stimulation trials by dividing it by the calculated mean EMG amplitude. In this manner, each participant's EMG during vestibular trials from a given session was scaled to the baseline EMG measured during that session. EMG data were high-pass filtered (30 Hz, zero lag, sixth-order Butterworth) and full-wave rectified. Data were cut into segments of 2048 data points (~1 s) providing a frequency resolution of ~0.98 Hz. For each participant and condition (e.g., pre-learning 100 ms delay), data were concatenated over the four balance trials providing 76 segments equating to 77.82 s of data. Over each segment (no overlap between segments), the autospectra for EVS and soleus EMG as well as the cross-spectra between EVS and EMG were calculated. The spectra were then averaged in the frequency domain to estimate coherence, gain, and cross-covariance. Coherence is a measure of the linear relationship between two signals in the frequency domain and is given by $C(f) = |P_{sr}(f)|^2 / (P_{ss}(f) P_{rr}(f))$ where $P_{sr}(f)$ is the stimulus-response cross spectrum, $P_{ss}(f)$ is the autospectrum of the EVS stimulus, and $P_{rr}(f)$ is the autospectrum of the rectified EMG. At each frequency point, coherence ranges from 0 (no linear relationship) to 1 (noise-free linear relationship). We interpreted coherence at each frequency point as significant if it exceeded the 95 % confidence limit derived from the number of disjoint segments (*Halliday et al., 1995*). Gain was computed by dividing the EVS-EMG cross spectrum by the EVS autospectrum and represents the ratio of the output signal to the input signal. Gain must be assessed alongside coherence because its estimate is unreliable at frequency points where coherence is below the significance threshold.

In the time domain, we estimated the non-normalized cross-covariance, which provides a time-domain measure of EVS-EMG association. We estimated cross-covariance for individual participants and group pooled data by taking the inverse Fourier transform of the EVS-EMG cross spectra (*Halliday et al., 1995*). Cross-covariance (sometimes referred to as cumulant density) estimates are used in neurophysiological and motor control studies to characterize the correlation between two signals (*Brown et al., 1999*; *Brown et al., 2001*; *Halliday et al., 1995*; *Halliday et al., 2006*; *Hansen et al., 2005*; *Nielsen et al., 2005*; *Tijssen et al., 2000*). These measures have been widely used to estimate

time domain vestibulo-motor responses to EVS during postural control activities (*Dakin et al., 2010*; *Dakin et al., 2007*; *Mackenzie and Reynolds, 2018*; *Mian and Day, 2009*; *Mian and Day, 2014*; *Reynolds, 2011*). During standing balance, lower-limb EVS-EMG cross-covariance responses exhibit a biphasic pattern with opposite peaks, defined as short- (50–70 ms) and medium-latency (100–120 ms) responses that match the responses elicited by square wave stimuli (*Dakin et al., 2010*; *Dakin et al., 2007*). For statistical analysis in Experiment 2- vestibular testing, the peak-to-peak amplitude of the cross-covariance estimates was extracted from each participant's response and used as a measure of response magnitude. When either of the short- or medium-latency cross-covariance peaks did not surpass the 95 % confidence interval, the value of that peak was set to zero (*Dakin et al., 2007*; *Forbes et al., 2016*). Therefore, if both short- and medium-latency peaks did not exceed the confidence intervals, the cross-covariance vestibular response amplitude was zero and considered absent.

For Experiment 3, we tracked time-varying changes in the vestibular-evoked muscle responses during transitions to 200 ms delayed balance control by estimating the time-varying coherence and gain between EVS and rectified EMG activity. Segments of 24 s of data, including 8 s prior to and after the imposed delay, from each trial were used for analysis. Because estimates of vestibulomuscular coherence and gain required multiple repetitions of the imposed delay, we did not evaluate vestibular responses on a trial-by-trial basis. Time–frequency vestibulomuscular coherence and gain were calculated by convolving the input EVS and output EMG with a set of complex Morlet wavelets (*Blouin et al., 2011*; *Zhan et al., 2006*), defined as complex sine waves tapered by a Gaussian (*Cohen, 2014*; *Cohen, 2019*). The peak frequencies of the wavelets were linearly spaced from 0.5 to 25 Hz in 40 steps, and the number of cycles used for each wavelet was logarithmically spaced from 3 to 12 with increasing wavelet peak frequency. This corresponds to a full-width at half maximum (FWHM) ranging from 168 ms to 2249 ms and a spectral FWHM range of 0.39–5.25 Hz (*Cohen, 2019*). For each participant, the wavelet analysis was performed on a data set consisting of all segments of data (each 24 s in length) and averaged across the number of transitions. To compare vestibular responses with the reported perception of unexpected balance, we only included transitions that were perceived by the participants (group data: 489 out of 584). A pooled wavelet analysis was also performed on a concatenated data set of all 489 participant data segments. Time–frequency contour plots of coherence and gain are presented with the first and last 2 s of data removed due to the distortion by window edge effects when applying wavelet analysis. For illustrative purposes, and because gain is only reliable when coherence is significant, we plotted non-significant coherence points and their gain counterparts as zero in the time-frequency contour plots (*Figure 6B*). To compare the overall strength of the EVS-EMG relationship during the period of baseline balance control vs. delayed balance control, we computed mean coherence and mean gain (*Luu et al., 2012*). Mean coherence and mean gain were calculated by averaging values across the 0.5–25 Hz bandwidth at each time point. For the mean gain estimate, only gain values for which coherence was significant at each corresponding time and frequency points were used (which was primarily across the 0–10 Hz bandwidth). To characterize the time course of vestibular response attenuation induced by the imposed delay, we fit exponential decay functions to each participant's average coherence and during the 8 s period of 200 ms delay. For gain, we fit an exponential only to the group mean gain because for some participants coherence decreased below significance at all frequencies for some periods of the delay exposure (see *Figure 6C*, left-lower panel).

## Psychometric functions

For perception trials, psychometric functions were generated using the participant button switch responses during the experimental delay periods. The following criteria were used to classify a participant's response: (1) detected, identified if the button was pressed at any time during an 8 s delay period; and (2) missed, identified if the button was not pressed during an 8 s delay period. If the button was pressed prior to the onset of the delay and held until the 8 s delay period started, the trial was removed from analysis (see *Table 3*). Mean detection rates for each delay were computed by dividing the number of detected balance control transitions (delay trial presented with button switch on) by the total number of used delay trials (see *Table 3*). Detection time was computed as the time between the onset of the delay and the button press. Psychometric functions were generated by relating the participant's proportion of detected responses to the magnitude of the delay (ms) to estimate a delay threshold level of detected unexpected balance behavior induced by the simulation

delays. Using custom software (LabVIEW 2013, National Instruments; *Rasman et al., 2021*; available at https://doi.org/10.5683/SP2/IKX9ML), a sigmoidal cumulative normal function was fit to each participant's response data across the different delays using a robust Bayesian curve fitting procedure (*Peters et al., 2016*; *Peters et al., 2015*). Briefly, we parameterized each participant's psychometric function using the following mixture model:

$$p\left(data|\mu, \sigma, \delta\right) = \frac{\delta}{2} + (1 - \delta)\left(\int_{-\infty}^{x} \frac{1}{\sqrt{2\pi\sigma^2}} e^{-\frac{(u-\mu)^2}{2\sigma^2}}\right)$$

where $\mu$ is the position of the sigmoidal curve along the x-axis, $\sigma$ is the slope of the sigmoidal curve, and $\delta$ is the realistic probability of a lapse in performance (e.g., loss of concentration, accidental button pushes, etc.) on some small proportion of trials (*Goldreich et al., 2009*; *Kontsevich and Tyler, 1999*). We set the range of possible $\mu$ values from 0 to 0.5 (seconds) in steps of 0.005, $\sigma$ (standard deviation) values from 0.01 to 5 in steps of 0.01, and $\delta$ (lapse rate) values from 0.01 to 0.05 (%) in steps of 0.01. We began with a uniform prior distribution over this parameter space. We marginalized over the $\delta$, a nuisance parameter in this case, to obtain a joint posterior probability distribution over $\mu$ and $\sigma$. The best-fitting curve was taken as the mode of this joint posterior probability distribution. Participants' perceptual threshold was calculated by averaging the interpolated 70 % correct threshold value across the joint posterior probability distribution.

For Experiment 3, we used only a 200 ms experimental delay because we were primarily interested in comparing perceptual detection times with vestibular response attenuation and relating those latencies to standing behavior (sway velocity variance). Because only one stimulus level (200 ms) was presented, we were unable to use psychometric functions to estimate a perceptual threshold and instead we simply report the proportion of 200 ms transitions that were detected.

## Statistical analysis

All statistical analyses were performed using SPSS22 software (IBM) and the significance level was set at 0.05. Group data in text, tables, and figures are presented as mean ± standard deviations unless otherwise specified.

### Experiment 1

To test our hypothesis that novel delays would influence standing balance behavior, we assessed changes in whole-body sway across delays using linear mixed models (fixed effect: delay level; random effect: participant ID). This analysis was run using the extracted sway velocity variance (from 2 s windows) and the percentage of the trial within the balance limits as dependent variables.

### Experiment 2

To establish how balance behavior changed during the 400 ms delay training session, we fit a first-order exponential function to the velocity variance data (2 s windows) obtained from all participants over the 100 min of training. We further compared differences in behavior at different time points using t-tests. Specifically, we assessed (1) whether balancing with the 400 ms delay after training was different than baseline balance by comparing the last minute of training to baseline standing, (2) adaptation during training by comparing the last minute of training to the first minute of training, and (3) whether the improvements in balance behavior were maintained after 3 months by comparing the first minute of retention to the first minute of training and the first minute of retention to the last minute of training. We performed identical analyses using the percentage of the trial within the balance limits as the dependent variable.

To evaluate how imposed delays influenced the vestibular-evoked balance responses (vestibular testing) and perception of standing motion (perceptual testing), we emphasized mainly on interactions (delay × learning) and main effects (delay, learning). Specifically, if there was an interaction, we quantified how the training protocol influenced the vestibular-evoked responses and perception of balance at each delay. Therefore, we decomposed any detected interaction effects by comparing pre-learning to post-learning and pre-learning to retention across delays. Accordingly, we performed planned comparisons with either paired t-tests (for parametric tests) or Wilcoxon signed-rank tests (non-parametric tests), which were Bonferroni corrected for multiple comparisons.

## Vestibular testing trials

To test our hypotheses that vestibular-evoked responses would first attenuate with imposed delays and then increase following the training protocol, we evaluated EVS-EMG cross-covariance responses across conditions. As there were missing data (only seven retention participants) and the responses were not normally distributed (Shapiro–Wilk test), we assessed the effects of delay and learning on the amplitude of vestibular-evoked cross-covariance (peak-to-peak) responses using a non-parametric analysis. We rank transformed the data and ran an ordinal logistic regression through the generalized estimated equations in SPSS (within-subject variables: delay and learning; participant variable: participant ID), which is a nonparametric test that accounts for repeated measures and missing data. We then decomposed the main effects and interaction effects using Bonferroni corrected Wilcoxon signed-rank tests. Because standing balance variability is partially linked to balance control errors (*Kiemel et al., 2002*) that may be evoked by the imposed delays, we also compared sway velocity variance using a linear mixed model (fixed effects: delay level and learning condition; random effect: participant ID) and decomposed the main and interaction effects using Bonferroni corrected pairwise comparisons.

## Perceptual testing trials

To test our hypothesis that sensorimotor adaptation to delayed balance control would reduce perceptual sensitivity to manipulated delays (i.e., shift psychometric functions to the right), we compared 70 % detection thresholds using a linear mixed model (fixed effect: learning condition; random effect: participant ID). Significant main effects of learning were decomposed using Bonferroni corrected pairwise comparisons. Additionally, to determine how balance control errors (see above) varied across conditions, we compared sway velocity variance during the delay period for perception trials using linear mixed models (fixed effects: delay level and learning condition; random effect: participant ID). We decomposed detected main and interaction effects using Bonferroni corrected pairwise comparisons.

## Experiment 3: vestibular and perception modulation

Experiment 3 was designed to track time-dependent modulations in the vestibular control of balance together with changes in sway velocity variance and perceptual detections. For these results, we report only descriptive statistics. For vestibular responses, we report the time when the vestibular-evoked muscle responses (coherence and gain) were attenuated by 63.2 % and 95 % (extracted from exponential decay functions) during the delay period. For perception, we report the group average perceptual detection time and 95th percentile. Finally, to link the vestibular and perceptual responses, we report the amount of vestibular attenuation that aligned with the average perceptual detection time.

## Acknowledgements

We thank Hasrit Sidhu for his help with data collection and all the participants who participated in this research. This study was funded by the Natural Sciences and Engineering Research Council of Canada, grant number RGPIN-2020-05438, awarded to J-SB. BGR received graduate student funding from the Natural Sciences and Engineering Research Council of Canada and The University of Otago (Post-graduate Research Scholarship). PAF received funding from the Netherlands Organization for Scientific Research (NWO #016. Veni. 188.049). RMP was funded by a Natural Sciences and Engineering Research Council Grant to JTI.

# Additional information

## Funding

| Funder | Grant reference number | Author |
|---|---|---|
| Natural Sciences and Engineering Research Council of Canada | Graduate Student Scholarship | Brandon G Rasman |
| Nederlandse Organisatie voor Wetenschappelijk Onderzoek | NWO #016. Veni. 188.049 | Patrick A Forbes |
| Natural Sciences and Engineering Research Council of Canada | RGPIN-2020-05438 | Jean-Sébastien Blouin |
| University of Otago | Post-graduate Research Scholarship | Brandon G Rasman |

The funders had no role in study design, data collection and interpretation, or the decision to submit the work for publication.

## Author contributions

Brandon G Rasman, Conceptualization, Formal analysis, Investigation, Methodology, Project administration, Software, Validation, Writing – original draft, Writing – review and editing; Patrick A Forbes, Ryan M Peters, Conceptualization, Formal analysis, Methodology, Writing – review and editing; Oscar Ortiz, Investigation, Writing – review and editing; Ian Franks, Conceptualization, Writing – review and editing; J Timothy Inglis, Romeo Chua, Conceptualization, Methodology, Writing – review and editing; Jean-Sébastien Blouin, Conceptualization, Formal analysis, Investigation, Methodology, Project administration, Resources, Software, Supervision, Validation, Writing – review and editing

## Author ORCIDs

Brandon G Rasman  http://orcid.org/0000-0002-8031-8320
Patrick A Forbes  http://orcid.org/0000-0002-0230-9971
Jean-Sébastien Blouin  http://orcid.org/0000-0003-0046-4051

## Ethics

Human subjects: The experimental protocol was verbally explained before the experiment and written informed consent was obtained. The experiments were approved by the University of British Columbia Human Research Ethics Committee and conformed to the Declaration of Helsinki, with the exception of registration to a database.

## Decision letter and Author response

Decision letter https://doi.org/10.7554/eLife.65085.sa1
Author response https://doi.org/10.7554/eLife.65085.sa2

# Additional files

## Supplementary files

• Transparent reporting form

## Data availability

We have created a Dataverse link for the source files needed to generate the group result figures. This can be found at https://doi.org/10.5683/SP2/IKX9ML.

The following dataset was generated:

| Author(s) | Year | Dataset title | Dataset URL | Database and Identifier |
|---|---|---|---|---|
| Rasman BG, Forbes PA, Peters RM, Ortiz O, Franks IM, Inglis JT, Chua R, Blouin JS | 2021 | Data and code for Learning to stand with unexpected sensorimotor delays | https://doi.org/10.5683/SP2/IKX9ML | Scholars Portal Dataverse, 10.5683/SP2/IKX9ML |

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
