## [Decision Letter]

**Acceptance summary:**

This is an exciting paper and provides new insights into sensorimotor learning for delay, including its time course, limits, and sensory mechanisms.

**Decision letter after peer review:**

Thank you for submitting your article "Learning to stand with unexpected sensorimotor delays" for consideration by *eLife*. Your article has been reviewed by 3 peer reviewers, including Noah J Cowan as Reviewer #1 and Reviewing Editor, and the evaluation has been overseen by Ronald Calabrese as the Senior Editor.

The reviewers have discussed their reviews with one another, and the Reviewing Editor has drafted this to help you prepare a revised submission. All reviewers agree there is the potential for this paper to provide a significant contribution to the literature, but there was strong agreement across the reviewers that there are fundamental gaps in the analysis and, potentially as a result, the interpretation of the results. In addition, each reviewer has read each others' reviews, and all three reviewers agree with the technical points of the other reviewers, and so each of these should be addressed in a revision.

Summary:

This manuscript will be of interest to a wide range of readers interested in motor control, motor adaptation and motor learning. Using an innovative robotic system, artificial time delays from muscle forces to movement were imposed during standing balance. The results clearly show both the difficulties that the imposed delays have on stabilizing movements and the ability of individuals to adapt to the imposed delays to maintain balance. This provides insight into how the nervous system is able to use sensory information to control movements under normal conditions given the time delays inherent in sensory, motor and neural processes.

Essential Revisions:

1. To characterize how strongly soleus EMG (the output) responds to vestibular stimulation (the input), standard measures designed to characterize input-output mappings should be used. Gain in the frequency domain and the unnormalized impulse response function in the time domain are the standard choices. Both measures used do not distinguish between decreases in EMG variability and gain.

1a) Coherence (while important to report) does not provide even an indirect measure of gain: two systems with gains different by an order of magnitude both reach maximum coherence of 1, depending on the amount of noise in the system. In terms of this research result, if an imposed delay increases output variance but does not change gain, then coherence decreases. Presumably soleus EMG variance increases with sway variance, so the decrease in coherence and Figure 6, for example, may simply reflect an increase in soleus EMG variance, not a decrease in the magnitude of vestibular-evoked muscles EMG responses.

2a) The authors use normalized cumulant density in the time domain as a stand in for gain. In principle, an unnormalized version of this should work, as it is related to the impulse response function (but not quite). It would be better to just directly use the standard estimate based on cross spectrum divided by the power spectrum of the input, rather than the normalized IFT of the cross spectrum that the authors used. But, by normalizing by the EMG signal properties (also by electrical stim -- but that would be the same/similar for all subjects), if EMG variance is increased with increasing delay (a plausible result), the cumulant density function amplitude would decrease due to this normalization, even if the vestibular gain remained unchanged.

So, in your revision, we require the use of a standard measure, e.g. cross spectral density between the input and output (as now calculated) divided by the power spectral density of the input. This would give a frequency domain response function. Taking the inverse Fourier transform would put this in the time domain as an estimate of the impulse response function. Other standard measures are possible.

Critically, this change in analysis may provide a new result, and thus require new interpretation. E.g., it may be that the "scalar" gain estimate does not change, but rather there is a change in the variance that explained your previous analysis. In this case, the transfer function estimate could be examined for alternative interpretations. The reviewers all felt that while this would not necessarily be as striking a result, it would still be nevertheless an important paper, and so it is important when re-analyzing this data not to have the forgone conclusion of a change in gain, which is not yet supported by the analysis.

2. We emphasize one of the major concerns of reviewer #2: Rather than report mean-removed RMS sway (in other words, sway standard deviation) it would be better to report percentage of time within specified sway limits

3. The authors should address whether existing models can stabilize an inverted pendulum with long feedback time delays. Sway variance will increase with increasing time delay, so at some point standing balance is not possible due to one's finite base of support (a type of plant nonlinearity). Specifically, the manuscript should be improved with a more nuanced Discussion (near lines 401-421) to consider the other classes of control models (optimal or intermittent control) that possibly could be candidates for controlling a system with long time delays. For example, if optimal control rather than proportional-derivative control is used, then it is possible in principle (perhaps excluding special cases) to stabilize a linear plant with arbitrarily long delays (e.g., Zhou and Wang 2014, DOI 10.1007/s10957-014-0532-8), although for long time delays such control can be quite fragile with respect to disturbances or plant parameters.

4. Bootstrapping should be better justified or not used.

*Reviewer #1:*

This paper aims to understand how imposed delays between ankle torque and whole-body motion destabilize standing balance, and determine the mechanisms that the nervous system uses to learn to compensate for these imposed delays. Related studies have estimated the critical delay at which upright standing is destabilized, and shown that humans can learn to compensate for increased delay. This study supports these prior results in the literature and delves deeper into human's standing balance. Specifically, they demonstrate that, although initially vestibulomotor responses are attenuated during imposed delays before learning, eventually subjects learn to partially recover their reliance on vestibular feedback, while, at the same time, improving postural control in the face of delay. Furthermore, after learning, subjects' ability to perceive unexpected delays was reduced, leading the authors to speculate that subjects may learn to "internalize" the experimentally added delay, making it hard to identify said delay an external perturbation. So, this study would be of broad interests for researchers who are studying human control of standing balance as well as other models of sensorimotor control across taxa.

Strengths:

1) The manuscript provides a clear and cogent motivation, and the hypotheses are well grounded in the literature.

2) The data, including the tables and graphs in this manuscript, verify the authors' hypotheses in a generally convincing way.

Weaknesses:

1) Clarification of velocity variance with bootstrapping: The estimates of sway velocity variance using bootstrapping was not well described nor justified. In figure 2, it said 'Sway velocity variance ..., and the resulting data were bootstrapped to provide a single estimate per participant and delay'. 'estimate per participant' sounds like simply using bootstrapping to estimate the mean, which is not a standard use of bootstrapping. It can be used to estimate SEM etc, also in line 795-797, it said 'we bootstrapped the participant's data with replication 10,000 times and then averaged across participants for each delay and learning condition.' from which it seems that the authors are computing the SEM across subjects. In sum the statistics associated with variance estimation and bootstrapping is unclear.

2) The use of Cumulant Density: cumulant density as a gain measure is not a standard technique for gain estimation in the motor control literature, and needs further justification.

3) The use of 2s windows for variance estimation in Experiment 1 was unclear. When estimating sway velocity variance in Experiment 1, 2s non-overlapping windows were applied and "if fewer than five 2 s windows were present, the average was taken across the available data (denoted with filled circles in figures)". And in Figure 2A, "Data not included in the velocity variance analysis are grayed out." However, some non-grayed-out regions in Fig 2A are not to be multiples of 2s, which is confusing.

Although the experimental methods, data analysis and writing skills in this paper are detailed, some improvements are needed for figure illustrations and terminology.

For example, Figure 1B is confusing. There is a circle with δ t inside which represents the imposed delay but it is not good to put both the "Whole body sway" and "Delayed whole-body sway" out of that circle. This is a confusing (and nonstandard!) way of schematizing a closed-loop system such as this. See many examples from the literature, including but not limited to papers by Lena Ting, Michael Dickinson, Daniel Robinson, my lab, and many others for standard ways of writing control system block diagrams that would be more easiliy interpretable. Lena Ting's lab (Lockhart and Ting 2007, expressed in time domain) and Simon Sponberg's lab (https://science.sciencemag.org/content/348/6240/1245.abstract, expressed as a general block diagram) both have nice papers that explicitly include delay in the feedback loop. My lab has three review papers with numerous examples that can be found here: https://limbs.lcsr.jhu.edu/publications/#Reviews.

Some figures in this paper are not straightforward for the readers to understand. Especially in Figure 1, the descriptions are complete, but figures themselves are not informative enough. For instance, in Figure 1 C, the descriptions of the experiment 3 whose trials ("of similar design to Experiment 2B, except that the robot only transitioned between baseline (20 ms) and 200 ms delays (not illustrated)") is unclear. This figure is significant to the paper's results in order to show the procedure of three experiments. Thus, it should be better clarified, or supplemental figures should be used to further explicate such issues.

More details about the load in Figure 1B is needed. We can only learn from line 541 that "For all experiments, participants stood on a custom-designed robotic balance simulator programmed with the mechanics of an inverted pendulum to replicate the load of the body during standing (Figure 1A)". However, the authors need to show in specific equations that were simulated, and the parameters they set in their design, to enable reproducibility of the results.

Moreover, important data points should be marked on the figure. On Figure 1E, the y axis "imposed delay" has only 20 ms and 300 ms marked, but authors later mentioned that "Data from a representative participant (see Figure 1E) show missed 299 detections of the 100 and 150 ms imposed delays", so it would be clearer if they can also mark 150 ms and 100 ms on the graph, because those are critical points.

RE: Cumulant Density: It is indeed critical to have a gain estimate in the vestibulomotor analysis, since coherence alone does not indicate the strength of response to a stimulus: nonlinearities and noise can decrease coherence, even if the gain to stimulus remains unchanged. The authors perform such a gain analysis (see e.g. Figure 4A) using "Cumulant density". This is an interesting technique for identifying the sensorimotor gain but does not seem to be in widespread use, and I could not find a theoretical justification in the system ID textbooks I have. That said, it appears possible that under the right conditions, the appropriately normalized Inverse Fourier Transform of the cross spectral density (i.e. the so-called Cumulant Density) should amount to the impulse response function, but I'm not sure of this. It would be helpful if the authors could better justify this approach or use a more standard analysis, such as impulse response recovery.

The authors call the Vestibular-Evoked Muscle Responses assays an "Experiment" (Experiment 2A), but the experiment is actually much broader, and covers all everything in the gray box in Figure 1C. Likewise for the perception assay ("Experiment 2B"). This terminology is confusing and I recommend dropping that nomenclature and just saying "Vestibular Testing" and "Perceptual Testing". Also, Figure 1C is laid out to look like a table, with two rows, and the first "column" appears to be labeling the rows, but it is not. Please rework this panel to be less confusing.

*Reviewer #2:*

Using a robotic system, the manuscript demonstrates that imposing time delays from ankle torques to movement causes postural sway to increase, as one would expect based on stochastic models of postural control. What is more surprising is the extent to which participants can adapt to imposed delays and decrease postural sway over multiple days. The evidence of adaptation is exceptionally clear. Another strength of the manuscript is that it relates these decreases in postural sway to a decrease in how often participants perceive unexpected balance movements, suggesting that over time participants learn that their ankles torques are causing the movements even though the movements are artificially delayed. These perceptual measures were obtained in real time during standing balance and carefully characterized, sometime that is not typically done in postural adaptation experiments.

The manuscript also characterizes the relationship between vestibular stimulation and surface electromyography EMG signals from the soleus muscle, using coherence in the frequency domain and the normalized cumulant density function in the time domain. Often gain, which has units of (output units)/(input units), is used to measure how responses to a perturbation change. A common example is using gain to quantify sensory re-weighting during standing balance. (It would be helpful if the authors discussed whether their hypothesized changes in vestibular-evoked muscle responses can be thought of as sensory re-weighting.)

One concern is whether the measures used in this study conflate changes in output magnitude with changes in output variance. For example, if an imposed delay increases output variance but does not change gain, then coherence decreases. For this reason, gain seems a better choice than coherence given the questions the authors are addressing. Similarly, in the time domain, an unnormalized impulse response function seems a better choice than the normalized cumulant density function. Similar comments apply to the time-frequency analysis of Experiment 3, which the manuscript uses to track changes in the relationship between vestibular stimulation and soleus EMG.

One important implication of the reported results that is discussed is that models that predict the maximal sensorimotor delay that allows standing balance are probably underestimating the maximal delay because they assume proportional-derivative (PD) or proportional-integral-derivative (PID) controllers, which are not optimal controllers when there are time delays. The predictions are also based a linear analysis of stability. In reality, due to a person's limited base of support, the person may fall or take a step even though a linear analysis shows that they are stable. It might be helpful to discuss this in relation to the study's imposition of specified limits of body sway.

One other issue that would be helpful to discuss in more detail would the exact sensory consequences of the imposed delays used in this study. The researchers imposed delays from the forces participants produce (forces applied to the support surface) to body movements. This delays sensory feedback related to body movements, but other sensory feedback, such as proprioceptive feedback about muscle forces and cutaneous feedback about the center of pressure (COP) have normal sensory delays. The component of COP dependent on center-of-mass position is delayed but the component dependent on ankle torque is not. Even the way proprioceptive information about muscle length is altered is more complex than a simple time delay, since length changes in tendons allow changes in muscles lengths without joint rotations. It would be helpful to discuss whether the resulting sensory conflicts contribute to the difficulty of balancing with an imposed delay and the implications for adaptation.

Abstract: It would be helpful to mention that sway was restricted to rotation about the ankles.

Figure 1 caption: Why was it necessary to use 3D googles to provide a visual scene to participants? Since the participants are actually moving, would it not be easier just to have participants look at an actual fixed visual scene?

Lines 171-174: Does "both p < 0.001" refer to overall tests for dependence on imposed delay? If so, it would be helpful to indicate which statements in this sentence are supported by statistical tests.

Lines 212-217: The comparing the retention test to sway attenuation corresponding to the time constant seems arbitrary and apparently does not take into account uncertainty in the estimated time constant. To support the statment that "balance improvements were partially maintained" it seems more relevant to compare the retention test to sway variance at the beginning of training to test whether improvements are at least partially maintained and to sway variance at the end of training to test whether improvements are only partially maintained.

Line 521: What was the range of participant ages?

Line 554: The delay from specified to measured motor position was estimated to be 20 ms. Is this with the inertial load of the backboard and participant? Should this be thought of a pure time delay or some other type of frequency response function (transfer function)?

Line 558: The linear least squares predictor algorithm used to synchronize the visual motion with the motors is not described in sufficient detail in the cited reference (Shepherd 2014) to permit others to reproduce this aspect of the experimental setup. It would be helpful to do so here.

Line 569: It would be helpful to specify the functional form and parameters of the stiffness that "caught" the backboard when it exceeded the specified limits.

Line 590: Does "reaching a limit" mean crossing from outside to inside the specified limits?

Lines 674-754: The text describing the protocols for Experiments 2A and 2B refer to pre-learning and post-learning testing, consistent with Figure 1C, but I cannot find where the details of these testing procedures are described. It would be helpful to explicitly refer to pre- and post-learning testing when describing these testing procedures in this section of the manuscript.

Line 697: Please clarify what it means that "Participants then completed Experiment 2A or 2B testing" during the retention testing.

Line 718: Was the range of root-mean-square amplitudes of the electrical vestibular stimulation due to stochastic variation or was amplitude systematically varied?

Line 712: Were conditions tested in order from small to large delays, as in Experiment 1?

Line 719: Does "pseudo-random order" refer to the order of trials within each condition?

Line 746: Was the transition from the 20-ms delay to the experimental delay instantaneous?

Line 746-750: It would be clearer to only use the term "transition" to refer to a change in imposed delay and use a different term to refer to the period of time during which the experimental delay is imposed.

Line 750: Please explain the reason a "catch" 20-ms delay was included in the experimental protocol, since this seems to be the same as the baseline delay. Did anything actually change in how the robotic system was controlled when the "catch" 20-ms delay was imposed? In other words, was the failure to find an effect preordained?

Line 776: Does the term "window" here mean the same thing as term "inter-transition delay" on Line 747? If so, it would be helpful to use the same term in both places.

Line 785: It is implied that the 2-s windows excluded periods of time when body angle was outside the specified limits, but it would be helpful to explicity state this restriction up front.

Line 795: What was bootstrapping used to estimate, the average variance for that participant? Why not just use the actual average variance, as was done when there were fewer than five 2-s windows? Bootstrapping is typically done to test a null hypthosis or construct a confidence interval and assumes independent samples, which would not be the case for data from the same participant.

Line 812: Did the 2-s window have to start after the delay was imposed AND end before the button press?

Line 821: Does concatenating the trials mean that the jumps in signal values from the end of one trial to the beginning of the next are affecting the results?

*Reviewer #3:*

The study describes three experiments that investigated the influence of increased feedback time delay in the ability of human subjects to maintain stable upright balance and their ability to learn/adapt to control balance at time delays values that are greater than those predicted to be possible based on existing simple feedback control models of balance. The studies made use of a unique robot balance device that allowed the generation of continuous body motion as a function of a time-delayed version of the corrective ankle torque generated by the subject as they swayed.

The experiments quantified changes in body sway behaviors as a function of added time delay and characterized the time course of improvements in balance control as subjects learned to stand with a long delay value imposed by the robotic device. The learning was acquired over multiple training sessions and was demonstrated both by a reduction over time in sway measures and by changes in psychophysical measures that demonstrated a reduction in a subject's indication of the occurrence of unexpected body motions. Additionally, the learned ability to control balance with an added time delay was very well retained at 3 months post training.

A final experiment used electrical vestibular stimulation (EVS) to demonstrate the changing contribution of vestibular information to balance control following a transient increase in time delay. This experiment demonstrated a marked reduction in activity in the soleus muscle that was correlated with the EVS following the onset of an added feedback time delay indicating that the added delay caused a reduction in the vestibular contribution to balance.

In general, the experiments were appropriately designed and analyzed. Although the number of participants in each experiment were not large, they were sufficient given the rather robust effects observed when time delays were added to be feedback.

The robotic balance device not only permitted manipulation of the time delay, but also guarded against subjects actually falling when they swayed beyond the normal range of sways compatible with stable balance. This artificial stabilization of balance beyond the normal range permitted subjects to recover from what would otherwise have been a fall and to immediately continue their attempts to learn a balance strategy that was able to overcome the detrimental effects of increased feedback delay. Thus the training procedure was likely much more effective than if trials had been stopped when sway moved beyond the normal range. One imagines such a training device could have important uses in rehabilitation of balance deficits.

But this artificial stabilization also interfered with one of the balance measures used to quantify the influence of added time delay. Specifically, the RMS sway measure used by the authors did not distinguish between time periods when the subject was being artificially stabilized and the time periods when sway was within the normal sway range. Thus the results that were based on the RMS sway measure were not convincing. But fortunately a sway velocity variance measure was also used to quantify sway behavior as a function of time delay and this measure only used data that was within the normal range of sway.

Overall results are an important addition to knowledge related to how humans control standing balance and demonstrate an ability to learn, with training, to balance with unexpectedly long delays between control action and the resulting body motion.

Overall results are clearly presented but this reviewer has some suggestions for improvement.

1. Experiment 1 quantifies changes in balance control as a function of added time delay by measuring RMS sway amplitude and sway velocity variance. It seems incorrect to use the entire 60 seconds of data in the calculation of RMS sway when there are periods of sway beyond the 3 deg backward and 6 deg forward sway balance limits since the backboard motion is artificially stabilized in the region outside the balance limits. It is only with detailed reading of the methods that the reader understands that the backboard motion is artificially stabilized at extremes of sway so this makes a complete understanding of the RMS sway results presented in Figure 2 difficult. But the problem is not fixed by just making the artificial stabilization methods more evident when presenting the Figure 2 results. The problem is that this RMS sway calculation is not representative of the overall subject-controlled sway behavior when there are periods in the trails when the sway was not controlled by the subject (which occurred frequently with longer delays). This reviewer suggests that a much more useful measure would be to calculate and display the percentage of time that subjects were maintaining control within the balance limits. It would be similarly informative to see this type of percentage measure used to track the Experiment 2 learning results in addition to the velocity variance measures shown in Figure 3.

2. In Experiment 3, group data results show changes in muscle activation evoked by electrical vestibular stimulation based on the 489 of 588 trials on which subjects signaled that they detected an unexpected balance motion. But what about the other 99 trials when subjects did not signal unexpected balance motion? Since the authors suggest there is a possible linkage between the time course of vestibular decline and the detection of unexpected motion, it may be that there was a slower (longer time constant) or reduced amplitude of vestibular decline on the 99 trials where unexpected motion was not detected. Such a finding would strengthen the notion of a linkage between vestibular inhibition and motion detection. Alternatively, if the vestibular declines were indistinguishable between trials with and without unexpected motion this would suggest that the linkage was not tight and something else may be involved. In either case this comparison would be useful. Additionally, it seems that 99 trials should be enough data since Figure 6 shows good results from a single subject based on 77 trials.

3. The section on pages 23 and 24 discusses alternative models that might be able to explain how subjects can learn to tolerate long time delays. I believe that all of the references to alternative models are to models that would be classified as continuous control models as opposed to intermittent control schemes that have been proposed as an alternative (e.g. Ian Loram references such as Loram et al., J Physiol 589.2:307-324, 2011, Gawthrop, Loram, Lakie, Biol Cybern 101:131-146, 2009, and the Morasso reference in the authors manuscript). It seems that Loram's work has shown that it is possible to visually control an unstable load with properties similar to those of a human body using an intermittent control scheme. This intermittent control scheme should be referenced. But beyond just mentioning these alternative control structures as possibilities, do the authors know of actual simulations of these models that can demonstrate that an inverted pendulum system can be made stable with the extremely long time delays that the authors investigated?

4. Several places in the manuscript the authors refer to estimates of maximal time delays based on simpler feedback control models. Specifically, the Bingham et al. 2011 and the van der Kooij and Peterka, 2011 references are given. But the values of the maximal time delays are not consistent across the various mentions of these two references. Here is a listing of those mentions:

- Line 78: ~300 ms

- Line 406: 340-430 ms

- Line 667: ~400 ms

- Line 678: ~400 ms

This reviewer could find mention of 340 ms in the van der Kooij and Peterka paper, but it seems that the Bingham paper did not really investigate time delay in detail and that paper also was investigating body motion in the frontal plane that has considerably different lower body dynamics compared to a single segment inverted pendulum.

5. The authors indicated that analysis programs and data will be made publicly available upon acceptance for publication.

---

## [Author Response]

Essential Revisions:1. To characterize how strongly soleus EMG (the output) responds to vestibular stimulation (the input), standard measures designed to characterize input-output mappings should be used. Gain in the frequency domain and the unnormalized impulse response function in the time domain are the standard choices. Both measures used do not distinguish between decreases in EMG variability and gain.1a) Coherence (while important to report) does not provide even an indirect measure of gain: two systems with gains different by an order of magnitude both reach maximum coherence of 1, depending on the amount of noise in the system. In terms of this research result, if an imposed delay increases output variance but does not change gain, then coherence decreases. Presumably soleus EMG variance increases with sway variance, so the decrease in coherence and Figure 6, for example, may simply reflect an increase in soleus EMG variance, not a decrease in the magnitude of vestibular-evoked muscles EMG responses.2a) The authors use normalized cumulant density in the time domain as a stand in for gain. In principle, an unnormalized version of this should work, as it is related to the impulse response function (but not quite). It would be better to just directly use the standard estimate based on cross spectrum divided by the power spectrum of the input, rather than the normalized IFT of the cross spectrum that the authors used. But, by normalizing by the EMG signal properties (also by electrical stim -- but that would be the same/similar for all subjects), if EMG variance is increased with increasing delay (a plausible result), the cumulant density function amplitude would decrease due to this normalization, even if the vestibular gain remained unchanged.So, in your revision, we require the use of a standard measure, e.g. cross spectral density between the input and output (as now calculated) divided by the power spectral density of the input. This would give a frequency domain response function. Taking the inverse Fourier transform would put this in the time domain as an estimate of the impulse response function. Other standard measures are possible.Critically, this change in analysis may provide a new result, and thus require new interpretation. E.g., it may be that the "scalar" gain estimate does not change, but rather there is a change in the variance that explained your previous analysis. In this case, the transfer function estimate could be examined for alternative interpretations. The reviewers all felt that while this would not necessarily be as striking a result, it would still be nevertheless an important paper, and so it is important when re-analyzing this data not to have the forgone conclusion of a change in gain, which is not yet supported by the analysis.

We thank the reviewers for highlighting these points regarding the analyses of vestibular-evoked muscle responses. Below we have provided answers to the points raised above.

For clarity, we have now replaced the term “cumulant density” with its equivalent, “cross-covariance” in the revised manuscript and the response to the reviewers. Cumulant density estimates are the equivalent of cross-covariance estimates and provide an analogous interpretation as cross-correlations (Halliday et al., 1995; Halliday et al. 2006). These measures are used often in neurophysiology and motor control studies to characterize the correlation between two signals (Brown et al., 1999; Brown et al., 2001; Halliday et al., 2006; Hansen et al., 2005; Nielsen et al., 2005). We also note that in our original submission, we estimated normalized cross-covariance (cumulant density) responses in order to compare vestibular-evoked muscle responses across the different testing sessions (pre-learning, post-learning, retention), which were performed on different days. This was to account for any inter-session variability in the surface EMG measurements, which can occur for example, due to changes in electrode placement or skin conductance. While we took precautions to ensure consistent EMG recordings across testing sessions (noting the electrode placement for each participant), there is still potential for inter-session variability in EMG signals. Therefore, to account for any differences in EMG measurements across sessions when using the non-normalized measures (i.e. cross-covariance and gain, see below), we first scaled the EMG signals by a baseline EMG measurement from each session prior to estimating the time and frequency domain measures. We outline this scaling approach and its effects on the data after the description of our modified vestibulomotor analyses (see *EMG scaling and* Response Figure 4).

We agree with the reviewers that our assessment of changes in vestibulomotor contributions to balance across different time delays could be better suited with non-normalized impulse response functions in the time domain and gain in the frequency domain. A summary of the changes made to the revised manuscript for both time and frequency domain measures is provided below.

Time-domain responses

As noted by the reviewers, one approach to estimate the impulse response function is to take the IFFT of the gain. The difficulty with this approach, however, is that gain estimates above 25 Hz become unreliable (i.e. coherence falls below significance) because we only applied a stimulus with power at frequencies up to and including 25 Hz. In fact, gains at frequencies > 25 Hz become erratic and very large because the cross-spectrum is divided by a very small (i.e., low power) autospectrum. Because taking the IFFT of the gain is performed using all frequencies, the resultant impulse response function is dominated by frequencies > 25 Hz and the estimate is very noisy (Author response image 1). This example analysis was performed on the pre-learning data (without scaling EMG) after concatenating the data from all subjects.

**Author response image 1. sa2fig1:** Pooled non-normalized cross-covariance (left) and impulse response (right) estimates from pre-learning data set sampled at 2000 Hz. Estimates were calculated by concatenating data from all eight participants.

An option to avoid these distortions is to first downsample the data to 50 Hz prior to estimating the impulse response functions. This approach eliminates the influence of these high frequencies but comes at the cost of resolution in the time domain. We performed this analysis (estimating both non-normalized cross-covariance and an impulse response function) on the pre-learning data after concatenating the data from all subjects. From these estimates (Author response image 2), we observed the typical biphasic response of the muscle to the electrical stimulation, as well as the delay-dependent decrease in the magnitude of the impulse response. This initial estimate suggests that the changes in the normalized cross-covariance from our original analysis were not simply dependent upon the variance of the measured EMG signals.

**Author response image 2. sa2fig2:** Pooled non-normalized cross-covariance (left) and impulse response (right) functions from pre-learning data, originally sampled at 2000Hz and then down-sampled offline to 50Hz. Note that cross-covariance and impulse response functions now provide similar results after down-sampled to 50Hz. Both responses, however, have poor resolution when compared to the original cross-covariance (Response Figure 1; 2000Hz).

To address the substantial loss in time-resolution, we then tried increasing the sampling frequency to 100 Hz (see Author response image 3). Despite only a moderate increase in the sampling rate, frequencies > 25 Hz dominated the response and again produced a very noisy impulse response function. Given these issues, our preferred approach is to follow the reviewers’ alternative suggestion of estimating the vestibular contributions across the different delay conditions using non-normalized cross-covariance estimates in the time domain. For Experiment 2, the results of this new analysis on the individual subject data revealed that the changes in non-normalized cross-covariance responses matched those of the normalized cross-covariance both across delays and conditions, as well as their interaction. These new data (and statistics) are presented in Figure 4, Tables 1 and 2 and pages 9-11 in the Results section of our revised submission.

**Author response image 3. sa2fig3:** Pooled non-normalized cross-covariance (left) and impulse response (right) functions from pre-learning data, originally sampled at 2000Hz and then down-sampled offline to 100Hz. Note that the impulse response functions already become noisy when the data are downsampled to 100Hz.

Frequency-domain responses

Estimating gain from our data was relatively straight-forward and these analyses have been added to the manuscript. Specifically, we now present both coherence and gain for estimating the vestibular-evoked muscle response in the frequency domain for Experiments 2 and 3. Gain must be assessed alongside coherence because its value is unreliable at frequency points where coherence is below the significance threshold. However, because coherence was below significance at delays ≥ 200 ms for the majority of frequencies, we limited the analysis of coherence and gain to qualitative assessments across delay using the pooled participant estimates of all subject (see Figure 4 in the revised manuscript). Overall, these pooled coherence and gain estimates followed the same trend as cross-covariance, decreasing as the time delay increased. These results provide additional evidence to support the conclusions drawn in our original submission.

EMG scaling

As noted above, to control for any inter-session variation in our EMG recordings, we first scaled the EMG signals by a baseline EMG measurement from each session. For each participant, soleus EMG from vestibular stimulation trials from a given session (e.g., pre-learning) was scaled using data from a baseline (20 ms delay on robot) quiet standing trial from the same day. In these baseline trials, participants stood quietly at their preferred posture (typically ~1 – 2° anterior). We estimated the mean EMG amplitude while participants maintained their preferred posture (± 0.25°). This baseline value was then used to scale EMG from vestibular stimulation trials performed on a given day. We then estimated vestibular-evoked muscle responses (coherence, gain, cross-covariance). We have added this information in the methods on pages 35-37.

To demonstrate the effects of this scaling here, we have estimated vestibular evoked muscle responses (cross-covariance) using three methods: normalized estimates (i.e., original submission), non-normalized estimates (without any scaling), and non-normalized estimates that were scaled by baseline EMG. Crucially, regardless of the approach, the main outcomes remain the same: vestibular-evoked responses attenuate with larger imposed delays (particularly in pre-learning) and partially return to normal amplitudes after training in post-learning and retention (see Author response image 4). Following the reviewers’ suggestion, in the revised manuscript we have presented the non-normalized responses (scaled to each session’s baseline EMG).

**Author response image 4. sa2fig4:** Pooled cross-covariance estimates of vestibular-evoked muscle responses. Normalized responses (left), non-normalized responses (center) and scaled non-normalized responses (right). All three approaches produce a similar outcome: vestibular-evoked response amplitudes attenuate with increasing imposed delays (particularly in pre-learning) and partially return to baseline levels after training (observed in post-learning and retention). Dashed lines are 95% confidence intervals.

2. We emphasize one of the major concerns of reviewer #2: Rather than report mean-removed RMS sway (in other words, sway standard deviation) it would be better to report percentage of time within specified sway limits

We appreciate the reviewers’ perspective that mean-removed RMS of sway – calculated over periods in which the participant can be within or outside the robotic simulation sway position limits – may not only reflect processes of balance since they include time periods of artificial stabilization. It was for this reason that we only provided this measure in Experiment 1. Our original aim was to compare this more common measure (Day et al. 1993; Hsu et al., 2007; van der Kooij et al., 2011; Winter et al., 2001) of balance behaviour to the alternative – though more appropriate – measure of sway velocity variance extracted over periods of balance within the specified sway limits. To avoid further confusion on this matter we removed the mean-removed RMS of sway measure from the paper. We further agree that the percentage of time a participant is balancing within the sway limits is a more useful measure and have added this analysis to the Methods and Results section when assessing Experiments 1 and 2 (training data).

3. The authors should address whether existing models can stabilize an inverted pendulum with long feedback time delays. Sway variance will increase with increasing time delay, so at some point standing balance is not possible due to one's finite base of support (a type of plant nonlinearity). Specifically, the manuscript should be improved with a more nuanced Discussion (near lines 401-421) to consider the other classes of control models (optimal or intermittent control) that possibly could be candidates for controlling a system with long time delays. For example, if optimal control rather than proportional-derivative control is used, then it is possible in principle (perhaps excluding special cases) to stabilize a linear plant with arbitrarily long delays (e.g., Zhou and Wang 2014, DOI 10.1007/s10957-014-0532-8), although for long time delays such control can be quite fragile with respect to disturbances or plant parameters.

We thank the reviewers for their suggestions. Specifically, in the Discussion section, “*learning to stand with novel sensorimotor delays”* (Lines 358 – 375), we have now included a discussion of different balance control models that may or may not be capable of stabilizing standing balance with long feedback time delays. Here, we discuss how feedback (i.e., PID) and optimal controllers are expected to perform with large delays. We note that Kuo (1995) has described the robustness of an optimal controller for standing balance to control delays, showing that at very small center of mass accelerations, the system can be stabilized with large (> 500 ms) delays. However, this robustness rapidly declines with increasing external disturbances (Kuo 1995), as is also expected from the paper the reviewers provided (Zhou and Wang 2014). Additionally, we considered both continuous and intermittent controllers which have been proposed for standing balance and discussed their potential performance in the context of standing with large delays. The nervous system may use a combination of continuous and intermittent controllers, as proposed from previous models (Elias et al., 2014; Insperger et al., 2015). Since we did not explicitly explore or test the performance of different models with balance delays in our study, we refrain from making extensive conclusions on the model that would be best suited to replicate our data.

4. Bootstrapping should be better justified or not used.

For simplicity, we have removed all bootstrapping analyses that were used in our estimation of sway velocity variance (Experiments 1 and 2). Importantly, this has not changed the main outcomes, i.e., the effects of delay and training on sway velocity variance (see Table 1, Figure 2, Figure 3, Figure 4, Figure 5).

Reviewer #1:This paper aims to understand how imposed delays between ankle torque and whole-body motion destabilize standing balance, and determine the mechanisms that the nervous system uses to learn to compensate for these imposed delays. Related studies have estimated the critical delay at which upright standing is destabilized, and shown that humans can learn to compensate for increased delay. This study supports these prior results in the literature and delves deeper into human's standing balance. Specifically, they demonstrate that, although initially vestibulomotor responses are attenuated during imposed delays before learning, eventually subjects learn to partially recover their reliance on vestibular feedback, while, at the same time, improving postural control in the face of delay. Furthermore, after learning, subjects' ability to perceive unexpected delays was reduced, leading the authors to speculate that subjects may learn to "internalize" the experimentally added delay, making it hard to identify said delay an external perturbation. So, this study would be of broad interests for researchers who are studying human control of standing balance as well as other models of sensorimotor control across taxa.Strengths:1) The manuscript provides a clear and cogent motivation, and the hypotheses are well grounded in the literature.2) The data, including the tables and graphs in this manuscript, verify the authors' hypotheses in a generally convincing way.Weaknesses:1) Clarification of velocity variance with bootstrapping: The estimates of sway velocity variance using bootstrapping was not well described nor justified. In figure 2, it said 'Sway velocity variance ..., and the resulting data were bootstrapped to provide a single estimate per participant and delay'. 'estimate per participant' sounds like simply using bootstrapping to estimate the mean, which is not a standard use of bootstrapping. It can be used to estimate SEM etc, also in line 795-797, it said 'we bootstrapped the participant's data with replication 10,000 times and then averaged across participants for each delay and learning condition.' from which it seems that the authors are computing the SEM across subjects. In sum the statistics associated with variance estimation and bootstrapping is unclear.

We thank the reviewer for raising these concerns regarding the bootstrapping of the sway velocity variance data. As recommended from all reviewers, we have removed all bootstrapping of sway velocity variance from our analyses. For Experiments 1 and 2, on a participant-by-participant basis, we averaged the sway velocity variance from the extracted 2-s windows to provide an estimate of sway velocity variance for each participant at each condition (e.g., Experiment 1, 200 ms delay). The procedure was the same as in our original submission except we did not perform any bootstrapping. We then performed the same statistical tests on the group data. Importantly, re-performing our analysis without any bootstrapping did not influence the outcomes for sway velocity variance in Experiments 1 and 2. In addition, for Experiment 1 and 2, we have also added an outcome measure – the percent of time participants balanced within the sway position limits – as recommended by the reviewers.

2) The use of Cumulant Density: cumulant density as a gain measure is not a standard technique for gain estimation in the motor control literature, and needs further justification.

We thank the reviewer for raising these points, and refer to our main response above regarding the re-analysis of vestibular-evoked muscle responses. Briefly, cumulant density estimates are the equivalent of cross-covariance estimates (interchangeable in the literature), and provide an analogous interpretation as a cross-correlation (Halliday et al. 1995; Halliday et al. 2006). These measures are used regularly in neurophysiology and motor control studies to characterize the correlation between two signals (Brown et al. 1999; Brown et al. 2001; Halliday et al. 1995; Halliday et al. 2006; Hansen et al. 2005; Nielsen et al. 2005; Tijssen et al. 2000) and have been used extensively to estimate vestibular-evoked responses during standing balance (Dakin et al. 2010; Dakin et al. 2007; Mackenzie and Reynolds 2018; Mian and Day 2009; 2014; Reynolds 2011). For clarity, we have now replaced the “cumulant density” with its equivalent, “cross-covariance”, which is the more widely recognized terminology.

Briefly describing our revisions for vestibular-evoked muscle responses, we have re-analyzed our data using non-normalized cross-covariance estimates for an estimation of the response in the time domain. While we performed an impulse response function analysis, we do not believe this measure is appropriate for our data and demonstrate this point in our response above. This was primarily because useful impulse response functions could only be estimated by downsampling the original data from 2000Hz to 50Hz. Due to the loss in time-resolution, however, our preferred approach is to follow the reviewer’s alternative suggestion of estimating the vestibular contributions across the different delay conditions using non-normalized cross-covariance estimates in the time domain and gain estimates in the frequency domain (at frequencies where coherence is above significance).

We have also added text in the Materials and Methods section (lines 827 – 875) of our revised manuscript, explaining and rationalizing our use of coherence, gain and cross-covariance to estimate the vestibular-evoked muscle response. The revised text reads

“To estimate the presence and magnitude of vestibular-evoked muscle responses in Experiment 2 vestibular testing, we used a Fourier analysis to estimate the relationship between vestibular stimuli and muscle activity in the frequency and time domains. […] Therefore, if both short and medium latency peaks did not exceed the confidence intervals, the cross-covariance vestibular response amplitude was zero and considered absent.”

3) The use of 2s windows for variance estimation in Experiment 1 was unclear. When estimating sway velocity variance in Experiment 1, 2s non-overlapping windows were applied and "if fewer than five 2 s windows were present, the average was taken across the available data (denoted with filled circles in figures)". And in Figure 2A, "Data not included in the velocity variance analysis are grayed out." However, some non-grayed-out regions in Fig 2A are not to be multiples of 2s, which is confusing.

We acknowledge that the description of the 2s data windows used to estimate sway velocity variance should have been clearer. Figure 2A highlights the regions of the trial that were within or outside the balance simulation limits (i.e., unshaded or gray shaded regions respectively) but that does not mean that every data point in the unshaded regions was used for the analysis. To clarify, we have added the following text to the methods section (Lines 791 – 797),

“Sway velocity variance was estimated only from data in which whole-body angular position was within the virtual balance limits (6° anterior and 3° posterior). Specifically, data were extracted in non-overlapping 2 s windows, when there was at least one period of 2 continuous seconds of balance within the virtual limits. Data windows were only extracted in multiples of 2 s. For instance, if there was an 11 s segment of continuous balance, we only extracted five 2 s windows (i.e., first 10 s of the segment).”

We have also clarified the description of Figure 2, stating “Sway velocity variance was calculated over 2 s windows (extracted by taking segments when sway was within balance limits for at least 2 continuous seconds)” (Lines 1261-1263).

Although the experimental methods, data analysis and writing skills in this paper are detailed, some improvements are needed for figure illustrations and terminology.For example, Figure 1B is confusing. There is a circle with δ t inside which represents the imposed delay but it is not good to put both the "Whole body sway" and "Delayed whole-body sway" out of that circle. This is a confusing (and nonstandard!) way of schematizing a closed-loop system such as this. See many examples from the literature, including but not limited to papers by Lena Ting, Michael Dickinson, Daniel Robinson, my lab, and many others for standard ways of writing control system block diagrams that would be more easiliy interpretable. Lena Ting's lab (Lockhart and Ting 2007, expressed in time domain) and Simon Sponberg's lab (https://science.sciencemag.org/content/348/6240/1245.abstract, expressed as a general block diagram) both have nice papers that explicitly include delay in the feedback loop. My lab has three review papers with numerous examples that can be found here: https://limbs.lcsr.jhu.edu/publications/#Reviews.

We agree with the reviewer and have modified the figure according to the reviewer’s suggestions. The original figure meant to depict that the participant was in control of the robot and that self-generated (participant) ankle torque could result in either normal robotic sway (20ms delay) or delayed robotic sway (> 20ms). The revised figure highlights that the participant is in the control loop and that the robotic simulation takes in measured ankle-produced torques and uses an inverted pendulum transfer function to output whole-body angular motion.

Some figures in this paper are not straightforward for the readers to understand. Especially in Figure 1, the descriptions are complete, but figures themselves are not informative enough. For instance, in Figure 1 C, the descriptions of the experiment 3 whose trials ("of similar design to Experiment 2B, except that the robot only transitioned between baseline (20 ms) and 200 ms delays (not illustrated)") is unclear. This figure is significant to the paper's results in order to show the procedure of three experiments. Thus, it should be better clarified, or supplemental figures should be used to further explicate such issues.

We thank the reviewer for raising this concern regarding Figure 1. We have made modifications to this figure, including changes to Figure 1B, 1C, and 1E. Figure 1B has now been updated to reflect a closed-loop feedback control and depicts the inverted pendulum transfer function used in the robotic simulation (see response to previous comment above). Figure 1C has been modified to reflect that Experiment 2 involved vestibular or perceptual testing in pre-learning, post-learning and retention sessions. Additionally, Figure 1C now states that Experiment 3 involved *“*simultaneous vestibular and perceptual testing with delays transitioning between 20 and 200 ms*”*. Figure 1E has been updated to mark the delays which were not perceived in this example trial (see response to comment below).

More details about the load in Figure 1B is needed. We can only learn from line 541 that "For all experiments, participants stood on a custom-designed robotic balance simulator programmed with the mechanics of an inverted pendulum to replicate the load of the body during standing (Figure 1A)". However, the authors need to show in specific equations that were simulated, and the parameters they set in their design, to enable reproducibility of the results.

We have now included more information regarding the equations used in the robotic balance simulation. This information has been added in the text (Lines 518-532). This text reads:

“For all experiments, participants stood on a custom-designed robotic balance simulator programmed with the mechanics of an inverted pendulum to replicate the load of the body during standing (Figure 1A). Specifically, the simulator used a continuous transfer function that was converted to a discrete-time equivalent for real-time implementation using the zero-order hold methodIθ¨+θ˙−mmgLθ= TθT=1Is2+s−0.971mgL

as described by Luu et al., (2011), where θ is the angular position of the body’s center of mass relative to the ankle joint from vertical and is positive for a plantar-flexed ankle position, T is the ankle torque applied to the body, mm is the participant’s effective moving mass, L is the distance from the body’s center of mass to the ankle joint, g is gravitational acceleration (9.81 m/s^2^), and I is mass moment of inertia of the body measured about the ankles mmL2. The body weight above the ankles was simulated by removing the approximate weight of the feet from the participant’s total body weight so that the effective mass was calculated as 0.971m, where m is the participant’s total mass.”

As mentioned above, we have modified Figure 1B to depict the inverted pendulum equation used in the robotic simulation.

Moreover, important data points should be marked on the figure. On Figure 1E, the y axis "imposed delay" has only 20 ms and 300 ms marked, but authors later mentioned that "Data from a representative participant (see Figure 1E) show missed 299 detections of the 100 and 150 ms imposed delays", so it would be clearer if they can also mark 150 ms and 100 ms on the graph, because those are critical points.

We have added the reviewer’s suggestions to Figure 1E.

RE: Cumulant Density: It is indeed critical to have a gain estimate in the vestibulomotor analysis, since coherence alone does not indicate the strength of response to a stimulus: nonlinearities and noise can decrease coherence, even if the gain to stimulus remains unchanged. The authors perform such a gain analysis (see e.g. Figure 4A) using "Cumulant density". This is an interesting technique for identifying the sensorimotor gain but does not seem to be in widespread use, and I could not find a theoretical justification in the system ID textbooks I have. That said, it appears possible that under the right conditions, the appropriately normalized Inverse Fourier Transform of the cross spectral density (i.e. the so-called Cumulant Density) should amount to the impulse response function, but I'm not sure of this. It would be helpful if the authors could better justify this approach or use a more standard analysis, such as impulse response recovery.

We thank the reviewer for raising these points and refer the reviewer to our main response above regarding the vestibular-evoked muscle response analysis. Briefly, we have re-analyzed our data using non-normalized cross-covariance (cumulant density) functions for an estimation of the response in the time domain. While we performed an impulse response function analysis, we do not believe this measure is appropriate for our data and demonstrate this point in our response above. We have also added a standard measure of gain to be presented alongside coherence in the frequency domain, as recommended by the reviewers. For Experiment 2, frequency domain measures (coherence and gain) were evaluated qualitatively using the pooled participant estimates because with delays ≥ 200 ms, single-participant coherence only exceeded significance at sporadic frequencies, and gain values are unreliable when coherence is not significant. Importantly, our main outcomes are confirmed with this re-analysis: increasing the imposed delay attenuates the vestibular response and training to stand with a 400 ms delay leads to these responses partially increasing.

The authors call the Vestibular-Evoked Muscle Responses assays an "Experiment" (Experiment 2A), but the experiment is actually much broader, and covers all everything in the gray box in Figure 1C. Likewise for the perception assay ("Experiment 2B"). This terminology is confusing and I recommend dropping that nomenclature and just saying "Vestibular Testing" and "Perceptual Testing". Also, Figure 1C is laid out to look like a table, with two rows, and the first "column" appears to be labeling the rows, but it is not. Please rework this panel to be less confusing.

We have reworked this part of the figure. As suggested, we have removed the Experiment 2A and 2B nomenclature and now state “vestibular testing” and “perceptual testing” for Experiment 2. These changes are reflected in the figure and throughout the manuscript text.

Reviewer #2:Using a robotic system, the manuscript demonstrates that imposing time delays from ankle torques to movement causes postural sway to increase, as one would expect based on stochastic models of postural control. What is more surprising is the extent to which participants can adapt to imposed delays and decrease postural sway over multiple days. The evidence of adaptation is exceptionally clear. Another strength of the manuscript is that it relates these decreases in postural sway to a decrease in how often participants perceive unexpected balance movements, suggesting that over time participants learn that their ankles torques are causing the movements even though the movements are artificially delayed. These perceptual measures were obtained in real time during standing balance and carefully characterized, sometime that is not typically done in postural adaptation experiments.The manuscript also characterizes the relationship between vestibular stimulation and surface electromyography EMG signals from the soleus muscle, using coherence in the frequency domain and the normalized cumulant density function in the time domain. Often gain, which has units of (output units)/(input units), is used to measure how responses to a perturbation change. A common example is using gain to quantify sensory re-weighting during standing balance. (It would be helpful if the authors discussed whether their hypothesized changes in vestibular-evoked muscle responses can be thought of as sensory re-weighting.)

The reviewer notes that the estimation of a gain function (in the frequency domain) is a common method to quantify how responses to a sensory perturbation change. In our revision, we have added a gain analysis to Experiment 2 and 3. We further elaborate on the addition of gain in our response to the reviewer’s next comment (below) and the group review major comments.

In response to the reviewer’s final point above that imposing delays in the sensorimotor control loop should lead to sensory feedback becoming less reliable, and consequently the associated sensorimotor feedback gains could decrease. This is predicted from computational models of standing balance that consider increasing sensory delays (Bingham et al. 2011; Le Mouel and Brette 2019; van der Kooij and Peterka 2011). Because our delays are imposed between ankle-produced torques and whole-body motion, we expect that visual, vestibular, and somatosensory cues of body motion should be delayed, while acknowledging that muscle spindle, GTO and cutaneous cues regarding muscle contractions remain unaltered. Therefore, from a sensory re-weighting perspective: visual, vestibular and somatosensory gains should decline. Although our results show that vestibular-evoked muscle response decrease, our experiments were not designed to assess whether remaining sensory channels were also modified in a manner that follows the principles of sensory re-weighting. Instead, our results show the partial return of vestibular response amplitude after training at a delay of 400 ms. These results suggest the involvement of alternative processes of sensorimotor recalibration, where the brain learns to associate delayed whole-body motion with self-generated motor commands. This is further supported by the changes in perception after training.

We have added some sentences in the discussion acknowledge the sensory re-weighting hypothesis in regards to our data. This discussion section now reads:

“The attenuation of vestibular-evoked responses to increased sensorimotor delays could be explained through processes of sensory re-weighting (Cenciarini and Peterka 2006; Peterka 2002), where the decreasing reliability in sensory cues when balancing with additional delays decreases feedback gains. […] Our results suggest that these forms of recalibration are possible for the vestibular control of standing balance and the perception of standing balance” (Lines 379-407).

One concern is whether the measures used in this study conflate changes in output magnitude with changes in output variance. For example, if an imposed delay increases output variance but does not change gain, then coherence decreases. For this reason, gain seems a better choice than coherence given the questions the authors are addressing. Similarly, in the time domain, an unnormalized impulse response function seems a better choice than the normalized cumulant density function. Similar comments apply to the time-frequency analysis of Experiment 3, which the manuscript uses to track changes in the relationship between vestibular stimulation and soleus EMG.

We acknowledge the reviewer’s concerns that adding a gain estimate to the analysis would be beneficial. We have added this analysis, and refer the reviewer to our main response to the group review comments. Briefly, we now present both coherence and gain for estimating the vestibular-evoked muscle response in the frequency domain for Experiments 2 and 3. Because gain is unreliable when coherence is not significant, we always present coherence and gain together. For Experiment 2, coherence and gain were evaluated qualitatively using the pooled participant estimates because for delays ≥ 200 ms single-participant coherence only exceeded significance at sporadic frequencies, and gain is unreliable at non-significant frequencies.

For our time domain estimates, we acknowledge the reviewer’s comments about normalized cumulant density responses. Briefly, we have replaced this estimate of vestibulomotor responses with non-normalized cross-covariance responses. To account for any differences across sessions (please see main response for details), we first scaled the EMG data by baseline muscle activity measured from the same session. In our main response, we also assessed the pros and cons of using cross-covariance vs impulse responses, and have chosen to cross-covariance as our primary outcome measure. Importantly, our results using non-normalized responses show the same main outcomes as with our previous analysis (normalized responses). Therefore, our observations of biphasic vestibular response attenuation with increased imposed delays are not simply due to an increase in output variance.

One important implication of the reported results that is discussed is that models that predict the maximal sensorimotor delay that allows standing balance are probably underestimating the maximal delay because they assume proportional-derivative (PD) or proportional-integral-derivative (PID) controllers, which are not optimal controllers when there are time delays. The predictions are also based a linear analysis of stability. In reality, due to a person's limited base of support, the person may fall or take a step even though a linear analysis shows that they are stable. It might be helpful to discuss this in relation to the study's imposition of specified limits of body sway.

The reviewer raises valid points regarding feedback control models (PD, PID) not being optimal in the presence of time delays and basing their predictions on linear analysis of stability. In our revised discussion, we have included a discussion of different control models and have incorporated these points raised by the reviewer, as well as those made by the other reviewers. These additions are included on lines 358-375, and reads:

“This remarkable ability for humans to adapt and maintain upright stance with delays raises questions regarding the principles underlying the neural control of balance. Compared to feedback controllers (i.e. proportional-derivative and proportional-integral-derivative), which are not optimal in the presence of delays, optimal controllers can model the control of human standing (Kiemel et al. 2002; Kuo 1995; 2005; van der Kooij et al. 1999; van der Kooij et al. 2001) and theoretically stabilize human standing with large (> 500 ms) delays (Kuo 1995). This ability, however, rapidly declines with increasing center of mass accelerations (Kuo 1995), including those driven by external disturbances (Zhou and Wang 2014). Although feedback and optimal controllers assume that the nervous system linearly and continuously modulates the balancing torques to stand (Fitzpatrick et al. 1996; Masani et al. 2006; van der Kooij and de Vlugt 2007; Vette et al. 2007), intermittent corrective balance actions (Asai et al. 2009; Bottaro et al. 2005; Gawthrop et al. 2011; Loram et al. 2011; Loram et al. 2005) may represent a solution when time delays rule out continuous control (Gawthrop and Wang 2006; Ronco et al. 1999). Intermittent muscle activations are also sufficient to stabilize the upright body during a robotic standing balance task similar to the one used in the present study (Huryn et al. 2014). The nervous system may use a combination of the controllers during standing (Elias et al. 2014; Insperger et al. 2015) but our study did not explicitly test for evidence of these different controllers or their ability to stabilize upright stance with large delays.”

In agreement with Reviewer 3, we speculate that by allowing participants to continue balancing after hitting these limits, learning may have been more effective as compared to halting the simulation every time sway moved beyond the normal range.

“The notable learning observed after training may have been partly due to participants being passively supported past the virtual balance limits because it prevented certain non-linear behaviors that disrupt continuous balance control such as taking steps or falls.” (Lines 354-357)

One other issue that would be helpful to discuss in more detail would the exact sensory consequences of the imposed delays used in this study. The researchers imposed delays from the forces participants produce (forces applied to the support surface) to body movements. This delays sensory feedback related to body movements, but other sensory feedback, such as proprioceptive feedback about muscle forces and cutaneous feedback about the center of pressure (COP) have normal sensory delays. The component of COP dependent on center-of-mass position is delayed but the component dependent on ankle torque is not. Even the way proprioceptive information about muscle length is altered is more complex than a simple time delay, since length changes in tendons allow changes in muscles lengths without joint rotations. It would be helpful to discuss whether the resulting sensory conflicts contribute to the difficulty of balancing with an imposed delay and the implications for adaptation.

The reviewer raises an important consideration. Indeed, our experimental set-up only allows for the manipulation of delays between the torques participants produced (measured from the force plate they stood on) and body movements (AP rotations about the ankle joint axis). Other feedback loops (i.e., between motor commands and sensory information related to muscle contractions) had normal delays. This could have resulted in sensory conflicts and potentially influenced balance stability with delays as well as the ability to adapt. We have added a section in the discussion titled – Limitations and other considerations – that discuss these possible implications. This discussion is found on lines 472-485 of the revised manuscript and reads:

“We manipulated the delay between ankle-produced torques (measured from the force plate) and the resulting whole-body motion (angular rotation about the ankle joints). This manipulation altered the timing between the net output of self-generated balance motor commands (i.e., ankle torques) and resulting sensory cues (visual, vestibular and somatosensory) encoding whole-body and ankle motion. However, the timing between motor commands and part of the somatosensory signals from muscles (muscle spindles and golgi tendon organs) and/or skin (cutaneous receptors under the feet) that are sensitive to muscle force (and related ankle torque) or movements and pressure distribution under the feet, were unaltered by the imposed delays to whole-body motion. This may have led to potential conflicts in the sensory coding of balance motion and may have influenced the ability to control and learn to stand with imposed delays. As methodologies to probe and manipulate the sensorimotor dynamics of standing improve, future experiments can be envisioned to replicate and modify specific aspects (i.e., specific sensory afferents) of the physiological code underlying standing balance. Such endeavors are needed to unravel the sensorimotor principles governing balance control.”

Abstract: It would be helpful to mention that sway was restricted to rotation about the ankles.

We have specified that the sway was restricted to rotation about the ankle joints in the abstract. Lines 40-42.

Figure 1 caption: Why was it necessary to use 3D googles to provide a visual scene to participants? Since the participants are actually moving, would it not be easier just to have participants look at an actual fixed visual scene?

For this experiment, yes, we could technically have the participants view a normal (or real) scene since they are moving. However, we instead took advantage of the systems available within the laboratory setup. The visual display system, with the use of 3D goggles, provides the participant with an immersive visual scene during the experiments. Because the screen was fixed next to the participant (they looked left to view it), the only alternative would be to have participants view the screen without a visual field leaving only a vague uniform grey display screen. We believe the virtual reality setup provided a natural visual environment.

Lines 171-174: Does "both p < 0.001" refer to overall tests for dependence on imposed delay? If so, it would be helpful to indicate which statements in this sentence are supported by statistical tests.

Yes, in the original submission this line referred to that there was a main effect of imposed delay on sway velocity variance and mean-removed RMS of sway position. We have modified this sentence and other statements to clarify which statistical tests reveal main effects. (Lines 128-147).

Lines 212-217: The comparing the retention test to sway attenuation corresponding to the time constant seems arbitrary and apparently does not take into account uncertainty in the estimated time constant. To support the statment that "balance improvements were partially maintained" it seems more relevant to compare the retention test to sway variance at the beginning of training to test whether improvements are at least partially maintained and to sway variance at the end of training to test whether improvements are only partially maintained.

Thank you for this comment. As suggested, we have provided these statistical comparisons for our two dependent variables of balance behaviour: sway velocity variance and for percent time within limits. This information has been added to the Results section on lines 179-192. The text reads:

“Sway velocity variance in the first minute of retention testing was ~60.8% lower than the sway velocity variance from the first minute of training (4.95 ± 2.32 [°/s]^2^ vs 12.62 ± 9.03 [°/s]^2^; independent samples t-test: t_(26)_ = -2.86, p < 0.01). Sway velocity variance at the first minute of retention testing, however, remained greater than the last minute of training (4.95 ± 2.32 [°/s]^2^ vs 2.55 ± 1.76 [°/s]^2^; independent samples t-test: t_(26)_ = 3.11, p < 0.01). Similarly, the first minute of retention was associated with a greater percentage of time within the balancing limits compared to the first minute of training (88 ± 9% vs 64 ± 9%; independent samples t-test: t_(26)_ = 6.67, p < 0.001), but less than the last minute of training (88 ± 9% vs 97 ± 3%; independent samples t-test: t_(26)_ = -3.68, p < 0.01). When using only data from participants who performed the retention session (n = 12; paired t-tests with df = 11), sway velocity variance and percent time within the balance limits revealed identical results (all p values < 0.01). Overall, these results indicate that while standing with an imposed 400 ms delay is initially difficult (if not impossible), participants learn to balance with the delay with sufficient training (i.e., > 30 mins) and this ability is partially retained three months later.”

Line 521: What was the range of participant ages?

The participants were 19-34 years of age. We have added this information (Lines 498-499).

Line 554: The delay from specified to measured motor position was estimated to be 20 ms. Is this with the inertial load of the backboard and participant? Should this be thought of a pure time delay or some other type of frequency response function (transfer function)?

This is a pure delay and is estimated from the motion command delivered to the motors and the movement of the motors.

Line 558: The linear least squares predictor algorithm used to synchronize the visual motion with the motors is not described in sufficient detail in the cited reference (Shepherd 2014) to permit others to reproduce this aspect of the experimental setup. It would be helpful to do so here.

We have added this requested information to the revised text. The text now reads:

“Rendering and projection of the visual scene took approximately 70 ms; therefore, a linear least-squares predictor algorithm was used to synchronize the visual motion (i.e., predict visual motion occurring 50 ms later) together with the motors at a delay of 20 ms (Shepherd 2014). The linear prediction model used 6 data points, and the coefficients were selected by fitting data of participant sway to the corresponding data shifted by the appropriate delay using a linear least squares method.” Lines 546-551.Line 569: It would be helpful to specify the functional form and parameters of the stiffness that "caught" the backboard when it exceeded the specified limits.

We have added the following text to the revised manuscript:

“To represent the physical limits of sway during standing balance, the backboard rotated in the AP direction about the participant’s ankles with virtual angular position limits of 6° anterior and 3° posterior (Luu et al., 2011; Shepherd 2014). When the backboard position exceeded these position limits, the program gradually increased the simulated stiffness such that the participants could not rotate further in that direction regardless of the ankle torques they produced. This was implemented by linearly increasing a passive supportive torque to a threshold equivalent to the participant’s body load over a rotation range of one degree beyond the balance limits (i.e., passively maintaining the body at that angle). Any active torque applied by participants in the opposite direction would enable them to get out of the limits. Finally, to avoid a hard stop at these secondary limits (i.e., 7° anterior and 4° posterior), the supportive torque was decreased according to an additional damping term that was chosen to ensure a smooth attenuation of motion.” Lines 556-567.

Line 590: Does "reaching a limit" mean crossing from outside to inside the specified limits?

Reaching or crossing a limit means the angular position of the backboard has exceeded the angular position limits (i.e. crossing from inside to outside the specified limits, and not outside to inside). We have clarified this on Lines: 586-587.

Lines 674-754: The text describing the protocols for Experiments 2A and 2B refer to pre-learning and post-learning testing, consistent with Figure 1C, but I cannot find where the details of these testing procedures are described. It would be helpful to explicitly refer to pre- and post-learning testing when describing these testing procedures in this section of the manuscript.

We have modified the text to provide clarity regarding Experiment 2 pre-learning and post-learning. Related to this, we have removed the nomenclature Experiments 2A and 2B (as suggested by Reviewer 1) and have replaced it with “vestibular testing” and “perceptual testing”. As the reviewer suggested, we have added text in this section of the manuscript to explain that the vestibular testing and perceptual testing procedures were performed in pre-learning, post-learning and retention sessions. This revised section also explains the testing procedures in details and can be found on lines 686–785.

Line 697: Please clarify what it means that "Participants then completed Experiment 2A or 2B testing" during the retention testing.

Participants for Experiment 2 either performed vestibular testing or perceptual testing. These testing sessions were performed prior to training (pre-learning), immediately after training (post-learning) or ~3 months after training (retention). We have removed nomenclature of Experiment 2A and 2B and replaced it with vestibular testing and perceptual testing, respectively. This is now explained in the Results and Materials and methods sections (193-264; 686–784). This has been updated in the manuscript text and figure descriptions.

Line 718: Was the range of root-mean-square amplitudes of the electrical vestibular stimulation due to stochastic variation or was amplitude systematically varied?

The range of RMS amplitudes of the electrical vestibular stimulation was indeed due to stochastic variation when generating the different stimulus profiles. We have included this additional information on lines 720-721.

Line 712: Were conditions tested in order from small to large delays, as in Experiment 1?

The delay conditions for Experiment 2 vestibular testing were randomly ordered (see lines 721-723).

Line 719: Does "pseudo-random order" refer to the order of trials within each condition?

For each testing session (pre-learning, post-learning, retention), participants performed 24 total trials (6 delay levels repeated 4 times each). The 24 total trials were completed in four subgroups, where each subgroup consisted of the six delay conditions which were randomly ordered. (Lines 721-723).

Line 746: Was the transition from the 20-ms delay to the experimental delay instantaneous?

Yes, the transition was instantaneous. We have added this information. (Line 751)

Line 746-750: It would be clearer to only use the term "transition" to refer to a change in imposed delay and use a different term to refer to the period of time during which the experimental delay is imposed.

We agree with the reviewer and have modified the text to refer to the change in the imposed delay value as the “transition” and the time the experimental delay is imposed as the “delay period.” These changes can be found on lines 747-784.

Line 750: Please explain the reason a "catch" 20-ms delay was included in the experimental protocol, since this seems to be the same as the baseline delay. Did anything actually change in how the robotic system was controlled when the "catch" 20-ms delay was imposed? In other words, was the failure to find an effect preordained?

During each trial, the robot transitioned instantaneously from baseline 20 ms balance delay to one of the experimental delays (20-350 ms) before returning to 20 ms. The inter-transition interval varied randomly between 7-10s. We included the catch 20ms delay to check whether participants were simply cueing off of an expectation of when unexpected behavior (i.e. standing with delays) would occur based on a prediction of the time period (i.e. the 7-10s inter-transition interval). Nothing changed in the robotic system control when the robot “transitioned” from baseline (20ms) to the catch 20ms delay. In other words, imposing of a catch trial amounted to: 7-10s of 20ms (inter-transition interval), followed by the 8s of 20ms catch, followed by another 7-10s of 20ms (inter-transition interval).

Line 776: Does the term "window" here mean the same thing as term "inter-transition delay" on Line 747? If so, it would be helpful to use the same term in both places.

We thank the reviewer for highlighting this confusion. We have changed the text to accurately describe these periods as inter-transition intervals, which is what was meant by the original term “window”. This change can be found on line 783.

Line 785: It is implied that the 2-s windows excluded periods of time when body angle was outside the specified limits, but it would be helpful to explicity state this restriction up front.

We have added text at the start of this section to explicitly state that we only extracted and analyzed data when whole-body angular position was within the simulated balance limits.

Specifically, the revised text reads:

“Sway velocity variance was estimated only from data in which whole-body angular position was within the virtual balance limits (6° anterior and 3° posterior). Specifically, data were extracted in non-overlapping 2 s windows, when there was at least one period of 2 continuous seconds of balance within the virtual limits. Data windows were only extracted in multiples of 2 s. For instance, if there was an 11 s segment of continuous balance, we only extracted five 2 s windows (i.e., first 10 s of the segment).” Lines 791 – 797.

Line 795: What was bootstrapping used to estimate, the average variance for that participant? Why not just use the actual average variance, as was done when there were fewer than five 2-s windows? Bootstrapping is typically done to test a null hypthosis or construct a confidence interval and assumes independent samples, which would not be the case for data from the same participant.

We have removed the bootstrapping analysis throughout the entire manuscript and instead, as suggested, use the actual velocity variance average.

Line 812: Did the 2-s window have to start after the delay was imposed AND end before the button press?

For Experiment 3, we identified peak sway velocity variance leading up to a perceptual detection. We used a 2-s sliding window to extract the time course of sway velocity variance. The 2-s sliding window of variance started 6s before the delay was imposed and ended 6s after the delay was imposed. We extracted the peak sway velocity variance between the start of the imposed delay and either the button press (for detections) or the end of the delay (for missed detections). (Lines 816-825; 1313-1316)

Line 821: Does concatenating the trials mean that the jumps in signal values from the end of one trial to the beginning of the next are affecting the results?

While concatenating the trials does result in jumps in the signal values (end of one trial to start of next trial), we avoided these jumps within the FFT analysis windows by segmenting the data carefully. Each of the vestibular trials was cut into 19 segments of 2048 data points (i.e., ~1 second segments), and the four vestibular trials with the same delay (i.e. pre-learning 100 ms) were then concatenated to provide 76 total segments. Auto and cross-spectra were then calculated from the individual segments and the spectra were averaged in the frequency domain. Therefore, there is no segment overlap in the FFT analysis for auto and cross-spectra. We provide details of this concatenation and state that there was no segment overlap in lines 840 – 846.

Reviewer #3:The study describes three experiments that investigated the influence of increased feedback time delay in the ability of human subjects to maintain stable upright balance and their ability to learn/adapt to control balance at time delays values that are greater than those predicted to be possible based on existing simple feedback control models of balance. The studies made use of a unique robot balance device that allowed the generation of continuous body motion as a function of a time-delayed version of the corrective ankle torque generated by the subject as they swayed.The experiments quantified changes in body sway behaviors as a function of added time delay and characterized the time course of improvements in balance control as subjects learned to stand with a long delay value imposed by the robotic device. The learning was acquired over multiple training sessions and was demonstrated both by a reduction over time in sway measures and by changes in psychophysical measures that demonstrated a reduction in a subject's indication of the occurrence of unexpected body motions. Additionally, the learned ability to control balance with an added time delay was very well retained at 3 months post training.A final experiment used electrical vestibular stimulation (EVS) to demonstrate the changing contribution of vestibular information to balance control following a transient increase in time delay. This experiment demonstrated a marked reduction in activity in the soleus muscle that was correlated with the EVS following the onset of an added feedback time delay indicating that the added delay caused a reduction in the vestibular contribution to balance.In general, the experiments were appropriately designed and analyzed. Although the number of participants in each experiment were not large, they were sufficient given the rather robust effects observed when time delays were added to be feedback.The robotic balance device not only permitted manipulation of the time delay, but also guarded against subjects actually falling when they swayed beyond the normal range of sways compatible with stable balance. This artificial stabilization of balance beyond the normal range permitted subjects to recover from what would otherwise have been a fall and to immediately continue their attempts to learn a balance strategy that was able to overcome the detrimental effects of increased feedback delay. Thus the training procedure was likely much more effective than if trials had been stopped when sway moved beyond the normal range. One imagines such a training device could have important uses in rehabilitation of balance deficits.But this artificial stabilization also interfered with one of the balance measures used to quantify the influence of added time delay. Specifically, the RMS sway measure used by the authors did not distinguish between time periods when the subject was being artificially stabilized and the time periods when sway was within the normal sway range. Thus the results that were based on the RMS sway measure were not convincing. But fortunately a sway velocity variance measure was also used to quantify sway behavior as a function of time delay and this measure only used data that was within the normal range of sway.

We agree with the reviewer’s point that interpreting mean-removed RMS of sway when the participant was beyond the simulation limits (and thus artificially stabilized by the robotic balance system) may not reflect processes of balance since they include time periods of artificial stabilization. It was for this reason that we provided this measure only in Experiment 1. Our original aim was to compare this more common measure of balance behaviour (Day et al. 1993; Hsu et al. 2007; Jeka et al. 2004; van der Kooij et al. 2011; Winter et al. 2001) to the alternative – though more appropriate – measure of sway velocity variance extracted over periods of balance within the specified sway limits. To avoid further confusion on this matter we removed the mean-removed RMS of sway measure from the paper. Furthermore, as the reviewer suggested, we have added the percent of time participants balanced within the position limits as an additional measure (see response to major comment below and manuscript revisions to Experiment 1 and 2 in the Methods and Materials and Results sections).

Overall results are an important addition to knowledge related to how humans control standing balance and demonstrate an ability to learn, with training, to balance with unexpectedly long delays between control action and the resulting body motion.

We thank the reviewer for the positive remarks about our study and its relevance.

Overall results are clearly presented but this reviewer has some suggestions for improvement.1. Experiment 1 quantifies changes in balance control as a function of added time delay by measuring RMS sway amplitude and sway velocity variance. It seems incorrect to use the entire 60 seconds of data in the calculation of RMS sway when there are periods of sway beyond the 3 deg backward and 6 deg forward sway balance limits since the backboard motion is artificially stabilized in the region outside the balance limits. It is only with detailed reading of the methods that the reader understands that the backboard motion is artificially stabilized at extremes of sway so this makes a complete understanding of the RMS sway results presented in Figure 2 difficult. But the problem is not fixed by just making the artificial stabilization methods more evident when presenting the Figure 2 results. The problem is that this RMS sway calculation is not representative of the overall subject-controlled sway behavior when there are periods in the trails when the sway was not controlled by the subject (which occurred frequently with longer delays). This reviewer suggests that a much more useful measure would be to calculate and display the percentage of time that subjects were maintaining control within the balance limits. It would be similarly informative to see this type of percentage measure used to track the Experiment 2 learning results in addition to the velocity variance measures shown in Figure 3.

The reviewer is correct that the mean-removed RMS of sway over the entire trial should not be interpreted as exclusively subject-controlled sway because for periods of balance where the limits are exceeded, the participants are artificially supported by the robot. It was for this reason that we (a) grayed out these time periods in the example sway data shown in Figure 2, and (b) extracted sway velocity variance using only periods of balance where the participant was within the limits. Because this distinction in the different balance measures is not entirely explicit when examining our results, we have agreed to remove the RMS sway measure and adopt the reviewer’s suggested improvement. Specifically, we now report the percentage of time participants maintained balance control within the limits alongside the sway velocity variance results for both for Experiment 1 (see Figure 2B) and the Experiment 2 training protocol (Figure 3B inset) results (pages 7-9). Briefly, these data demonstrate that for Experiment 1, the percent time within limits declined at delays ≥ 200 ms. For the Experiment 2 training data (balancing with a 400 ms delay), participants gradually increased the percent time they balanced within the limits, reaching ~97% (of 60s) by the final minute of training.

2. In Experiment 3, group data results show changes in muscle activation evoked by electrical vestibular stimulation based on the 489 of 588 trials on which subjects signaled that they detected an unexpected balance motion. But what about the other 99 trials when subjects did not signal unexpected balance motion? Since the authors suggest there is a possible linkage between the time course of vestibular decline and the detection of unexpected motion, it may be that there was a slower (longer time constant) or reduced amplitude of vestibular decline on the 99 trials where unexpected motion was not detected. Such a finding would strengthen the notion of a linkage between vestibular inhibition and motion detection. Alternatively, if the vestibular declines were indistinguishable between trials with and without unexpected motion this would suggest that the linkage was not tight and something else may be involved. In either case this comparison would be useful. Additionally, it seems that 99 trials should be enough data since Figure 6 shows good results from a single subject based on 77 trials.

The reviewer raises an interesting question, and one that we had briefly explored in our original data analysis. To address this question, we first extracted all transitions that were missed from all subject data together with an equal number of randomly selected transitions that were detected. For instance, for a subject who missed 13 transitions, we randomly selected 13 transitions that were detected. Therefore, each subject contributed the same number of missed and detected transitions to our final estimate of coherence and gain. In total, this procedure provided 82 missed transitions and 82 detected transitions. We then computed time-frequency coherence and gain using this subset of data according to the same approach employed for the total dataset. From these time-frequency estimates, mean coherence and mean gain were calculated across the 0-25 Hz bandwidth throughout the trial (Author response image 5). Finally, we also tracked the sway velocity variance (2s sliding window, as in our original analysis) for the missed and detected trials to examine any changes in balance behaviour.

**Author response image 5. sa2fig5:** Experiment 3 time-varying EVS-EMG coherence, gain and sway velocity variability during detected (perceived) and missed (not perceived) delay transitions. Data are presented across transition periods, where the simulation transitioned from baseline to 200 ms delayed balance control, which lasted for 8s (between dashed red vertical lines). Top panel, a comparison of 82 detected and missed transitions vestibular-evoked muscle coherence and gain. Dashed lines represent coherence and gain levels which are 2 standard deviations below the mean levels in the 6 seconds preceding the onset of the delay. Bottom panel, a comparison of sway velocity variance from 82 detected and 82 missed transitions. Thick lines represent the mean with lighter lines representing the s.e.m. For both panels, blue and red traces represent detected and missed transitions, respectively.

During the detected transitions, both coherence and gain decreased following the onset of the 200 msec delay and returned to normal levels when the delay was removed. Although limited attenuation of the mean coherence and gain was observed in the missed transitions, the variability in both missed and detected trials was large such that the difference between detected and missed transitions was not substantial. We also observed that sway velocity variance during the 82 missed transitions was smaller (reaching a peak ~3x lower) than the 82 detected transitions. As a result, we cannot be certain that the (limited) difference in coherence and gain between detected and missed transitions is related entirely to the perceptual motion detection, because the whole-body sway behavior (i.e., sway velocity variance) is clearly different. As mentioned in our original submission, and revised manuscript, our results from Experiment 2 and 3 suggest that whole-body sway behavior influences both vestibular muscle responses and perception of unexpected balance motion. Currently, we have only presented this analysis in the response to the reviewer’s document. If the reviewer feels strongly about including this in the manuscript, we would be happy to include it as a supplementary figure.

3. The section on pages 23 and 24 discusses alternative models that might be able to explain how subjects can learn to tolerate long time delays. I believe that all of the references to alternative models are to models that would be classified as continuous control models as opposed to intermittent control schemes that have been proposed as an alternative (e.g. Ian Loram references such as Loram et al., J Physiol 589.2:307-324, 2011, Gawthrop, Loram, Lakie, Biol Cybern 101:131-146, 2009, and the Morasso reference in the authors manuscript). It seems that Loram's work has shown that it is possible to visually control an unstable load with properties similar to those of a human body using an intermittent control scheme. This intermittent control scheme should be referenced. But beyond just mentioning these alternative control structures as possibilities, do the authors know of actual simulations of these models that can demonstrate that an inverted pendulum system can be made stable with the extremely long time delays that the authors investigated?

We thank the reviewer for raising these points regarding the discussion of balance control models that may or may not be able to stabilize upright stance with long delays. We refer to our response to all the reviewers (at the start of this document) for how we addressed these points. Briefly, we have now added a discussion of the different control models for standing balance and how they may (or may not) be capable of replicating our results of stabilizing human standing with large delays. This discussion can be found on lines 358-375.

To answer the reviewer’s question of model simulations that can stabilize with long delays, we note that Kuo (1995) has described the robustness of an optimal controller for standing balance to control delays, showing that at very small center of mass accelerations, the system can be stabilized with large (> 500 ms) delays. However, this robustness rapidly declines with increasing external disturbances (Kuo 1995) as is also expected from the paper the reviewers provided (Zhou and Wang 2014). We include these points in our discussion on lines 358-375.

4. Several places in the manuscript the authors refer to estimates of maximal time delays based on simpler feedback control models. Specifically, the Bingham et al. 2011 and the van der Kooij and Peterka, 2011 references are given. But the values of the maximal time delays are not consistent across the various mentions of these two references. Here is a listing of those mentions:- Line 78: ~300 ms- Line 406: 340-430 ms- Line 667: ~400 ms- Line 678: ~400 ms

We agree with the reviewer that this inconsistency should be rectified. Considering the reviewer’s next comment (that the Bingham et al., 2011 study focused on frontal plane standing control), we are removing the Bingham et al., (2011) study when referring to critical delays (although we include the study to discuss balance delays in general). Computational models of standing suggest that upright posture cannot be maintained with delays ~300-340 ms (Milton and Insperger 2019; van der Kooij and Peterka 2011). Considering that inherent sensorimotor loop delays in human stance can vary (~100-160ms), we consider that any imposed delay (via robotic simulation) above 200ms would therefore surpass the low end of the critical delay range. For consistency, we now refer to previously proposed critical delays to be in the range of ~300-340ms.

This reviewer could find mention of 340 ms in the van der Kooij and Peterka paper, but it seems that the Bingham paper did not really investigate time delay in detail and that paper also was investigating body motion in the frontal plane that has considerably different lower body dynamics compared to a single segment inverted pendulum.

The Bingham paper mentions a critical delay of 429 ms on page 441 of their manuscript and describes its calculation in the appendix section. That being said, we agree with the reviewer that because Bingham et al., (2011) were assessing frontal plane (ML) control dynamics, this critical delay may not be the same for sagittal plane (AP) control. Therefore, we have removed explicitly referencing Bingham’s estimated ~430 ms critical delay for this paper. Nevertheless, we still refer to the Bingham paper when discussing delays in balance control in general.

5. The authors indicated that analysis programs and data will be made publicly available upon acceptance for publication.

We have created a dataverse link for the source files and code needed to generate the group result figures. This can be found at https://doi.org/10.5683/SP2/IKX9ML.